# SFPQ and NONO suppress RNA:DNA-hybrid-related telomere instability

Eleonora Petti[1,2,6], Valentina Buemi[1,2], Antonina Zappone[1,2], Odessa Schillaci[1], Pamela Veneziano Broccia[1,2], Roberto Dinami[1,2,6], Silvia Matteoni [3], Roberta Benetti[4,5] & Stefan Schoeftner[1,2]

In vertebrates, the telomere repeat containing long, non-coding RNA TERRA is prone to form RNA:DNA hybrids at telomeres. This results in the formation of R-loop structures, replication stress and telomere instability, but also contributes to alternative lengthening of telomeres (ALT). Here, we identify the TERRA binding proteins NONO and SFPQ as novel regulators of RNA:DNA hybrid related telomere instability. NONO and SFPQ locate at telomeres and have a common role in suppressing RNA:DNA hybrids and replication defects at telomeres. NONO and SFPQ act as heterodimers to suppress fragility and homologous recombination at telomeres, respectively. Combining increased telomere fragility with unleashing telomere recombination upon NONO/SFPQ loss of function causes massive recombination events, involving 35% of telomeres in ALT cells. Our data identify the RNA binding proteins SFPQ and NONO as novel regulators at telomeres that collaborate to ensure telomere integrity by suppressing telomere fragility and homologous recombination triggered by RNA:DNA hybrids.

[1] Genomic Stability Unit, Laboratorio Nazionale—Consorzio Interuniversitario per le Biotecnologie (LNCIB), Padriciano 99, 34149 Trieste, Italy. [2] Department of Life Sciences, Università degli Studi di Trieste, Via E. Weiss 2, 34127 Trieste, Italy. [3] Cellular Networks and Molecular Therapeutic Targets, Proteomics Unit, IRCCS—Regina Elena National Cancer Institute, via Elio Chianesi 53, 00144 Rome, Italy. [4] Dipartimento di Area Medica (Dame), Università degli Studi di Udine, p.le Kolbe 1, 33100 Udine, Italy. [5] Cancer Epigenetics Unit, Laboratorio Nazionale—Consorzio Interuniversitario per le Biotecnologie (LNCIB), Padriciano 99, 34149 Trieste, Italy. [6] Present address: Oncogenomic and Epigenetic Unit, IRCCS—Regina Elena National Cancer Institute, via Elio Chianesi 53, 00144 Rome, Italy. These authors contributed equally: Eleonora Petti, Valentina Buemi, Antonina Zappone. Correspondence and requests for materials should be addressed to S.S. (email: stefan.schoeftner@lncib.it)

Telomeres are heterochromatic structures that protect eukaryotic chromosome ends from an unwanted elicitation of a DNA damage response, telomere degradation and aberrant recombination[1–3]. Incomplete replication of chromosome ends results in progressive shortening of telomeres, finally leading to telomere dysfunction and irreversible cell cycle arrest, also referred to as replicative senescence[4]. Core components of vertebrate telomeric chromatin comprise the multi-protein complex "shelterin" that has a key role in telomere protection but also TERRA, a telomere repeat [UUAGGG$_n$] containing long non-coding RNA[5,6]. In human cancer cells, DNA methylation sensitive promoters located in subtelomeres recruit RNA Polymerase II that uses the CCCTAA-repeat containing telomere strand as template for the transcription of TERRA[5–11]. TERRA is heterogeneous in size and a subfraction of TERRA has been shown to localize to telomeric chromatin[5,6,11]. Accordingly, TERRA is reported to act as scaffold that promotes the concentration of proteins or enzymatic activities at telomeres, thereby

**Fig. 1** SFPQ and NONO interact with TERRA and localize to telomere repeats. **a** Silver staining of SDS-PAGE gels of eluates obtained from RNA-pull-down experiments using biotinylated r[UUAGGG]₆, biotinylated r[EGFP] RNA oligonucleotides or empty beads. Candidate TERRA interacting proteins identified by mass spectrometry are indicated. NE, nuclear extract used as input; MW, molecular weight marker. **b, c** Western blotting analysis of RNA-pull-down eluates using specific anti-NONO and anti-SFPQ antibodies confirms binding specificity of NONO and SFPQ for UUAGGG RNA repeats in mESCs (**b**) and H1299 cells (**c**). FUS was previously reported to interact with TERRA[42]. Actin was used as loading control; NE, nuclear extract was used as input. Source blots are available as Supplementary Figure 6 and 7. **d** Representative image of co-localization events (arrowheads) between NONO and the shelterin protein TRF2 as determined by confocal microscopy in U-2 OS cells. **e** Representative image of co-localization events (arrowheads) between SFPQ and the shelterin protein TRF1 by confocal microscopy in U-2 OS cells. **f** Quantification of **d** and **e**. Mean number of NONO-TRF2 and SFPQ-TRF1 co-localization events per nucleus is shown. Error bars indicate standard deviation. $n$ = number of analyzed nuclei. **g** Chromatin immunoprecipitation experiments (ChIP) using U-2 OS cells and mouse anti-TRF2, rabbit anti-histone H3, rabbit anti-FUS, rabbit anti-NONO, and rabbit anti-SFPQ antibodies. Mouse and rabbit control IgGs (IgG M/IgG R) were used as negative control. Serial dilutions of chromatin extract (input) prepared from U-2 OS cells were loaded. **h** Quantification of three independent ChIP experiments, average enrichment of telomeric repeats is indicated; s.e.: short exposure; l.e.: long exposure. **f, h** N, number of independent experiments, whiskers indicate standard deviation; a two-tailed Student's $t$-test was used to calculate $p$-values. Source data is provided as a Supplementary Information File. Scale bar, 1 μm. siRNAs listed in Supplementary Table 1

impacting on chromatin structure, chromosome end protection, replication but also telomere maintenance by telomerase[12–14]. Importantly, due to its G-rich sequence content, TERRA is prone to form RNA:DNA hybrids with the C-rich telomeric strand. This results in the displacement of the TTAGGG repeat containing telomere strand, giving rise to so-called "R-loop" structures[15–19]. In a natural context, R-loops are relevant for the regulation of gene expression, chromatin structure and IgG class switch recombination[20–29]. However, R-loops can represent obstacles to DNA replication resulting in replicative stress, chromosome fragility, DNA lesions and the activation of recombination-mediated DNA repair that can finally lead to chromosome rearrangements[28,30–34]. Recent studies showed that several biological pathways prevent or resolve RNA:DNA-hybrid structures at eukaryotic telomeres. In yeast, loss of the 5′ to 3′ exonuclease, Rat1p was linked with increased abundance of telomeric RNA:DNA hybrids[35]. Loss of the THO complex in yeast results in TERRA accumulation at telomeres, R-loop formation and increased telomere recombination rates[16]. Further, lack of the flap endonuclease 1 (FEN1), a canonical lagging strand DNA replication protein, leads to replicative stress at telomeres, caused by increased telomeric RNA:DNA hybrids[36]. In yeast and human cells, a particular relevance has been attributed to RNaseH that antagonize R-loop formation by degrading RNA paired with DNA[15,17,35]. In yeast, loss of RNaseH function results increased abundance of telomeric R-loops, higher telomere recombination rates, and a delayed onset of replicative senescence in telomerase-negative cells[15]. Telomerase-negative human ALT cells, that maintain telomere function by homologous recombination-based "alternative lengthening of telomeres" (ALT), have been demonstrated to display an increased abundance of R-loops and RNaseH1 localization at telomeres[17]. Depletion of RNaseH1 in these cells results in enhanced R-loop formation, replicative stress, selective fragility at the telomeric CCCTAA-repeat containing strand and increased extra-chromosomal telomere repeat content[17]. This suggests that R-loop-related activation of recombination-based DNA repair mechanisms can translate into improved telomere repeat maintenance. This may be of special relevance in telomerase-negative tumors that represent approximately 10–15% of all human tumors[37,38].

Here, we set out to investigate novel pathways that impact on RNA:DNA-hybrid-related regulation of telomere function. We show that the TERRA RNA binding proteins NONO and SFPQ are novel protein components of telomeric chromatin that have a central role suppressing RNA:DNA-hybrid formation, DNA replication defects, recombination, and DNA damage at telomeres of telomerase-positive and telomerase-negative cancer cells. Remarkably, NONO and SFPQ antagonize different downstream effects of telomeric RNA:DNA hybrids: NONO suppresses telomere fragility; in contrast SFPQ represents a powerful barrier to homologous recombination at telomeres. Accordingly, combined loss of NONO and SFPQ results in a massive increase of telomere recombination events and rapid alterations in telomere length in both, telomerase-positive and negative cells. Our study introduces NONO and SFPQ as novel regulators of RNA:DNA-hybrid management that may open new inroads in defining strategies that aim to target recombination-based pathways of telomere maintenance in human cancer.

## Results

**NONO and SFPQ are TERRA interacting proteins.** In order to identify novel TERRA interacting proteins, we performed TERRA RNA-pull-down experiments using mouse embryonic stem cells that maintain telomeres via telomerase dependent and independent telomere maintenance pathways[39]. Biotinylated RNA oligonucleotides containing six UUAGGG repeats or control oligonucleotides were incubated with nuclear extracts obtained from mouse embryonic stem cells and recovered by using stretpavidine coated agarose beads. Eluates from TERRA and control RNA-pull-down experiments were processed by gel electrophoresis. Silver staining revealed a series of proteins that were specifically eluted from TERRA RNA oligonucleotides (Fig. 1a). Mass spectrometric analysis revealed the identity of a large set of reported TERRA interacting proteins, such as a series of hnRNPs, as well as Fused in Sarcoma (FUS)[40–42]. Importantly, the TERRA-pull-down approach identified the Non-POU domain-containing octamer-binding protein (NONO) and Splicing factor proline- and glutamine-rich (SFPQ) as TERRA interacting proteins that have not yet been studied in the context of telomere function. The specificity of SFPQ and NONO for TERRA was validated by western blotting using TERRA-pull-down eluates obtained from mouse embryonic stem cells or H1299 non-small cell lung cancer cells (Fig. 1b, c). NONO and SFPQ belong to the Drosophila behavior/human splicing (DBHS) family protein that are defined by highly conserved tandem N-terminal RNA recognition motifs (RRMs), a NonA/paraspeckle domain (NOPS) and a C-terminal coiled-coil[43]. NONO and SFPQ function is reported to depend on the obligatory formation of homo- or heterodimers of both proteins[43]. In line with this, immunoprecipitation experiments using extracts from telomerase-positive H1299 and telomerase-negative U-2 OS osteosarcoma cells that maintain telomeres via the recombination-based "alternative lengthening of telomeres" (ALT) pathway, confirm a direct interaction between NONO and SFPQ in the cell model systems used in this study (Supplementary Figure 1A, B). Confocal microscopy revealed that a significant fraction of nuclear restricted SFPQ and NONO foci

co-localize with telomere repeat binding factor 2 (TRF2) or telomere repeat binding factor 1 (TRF1) in U-2 OS interphase cells (Fig. 1d–f). Immunoprecipitation experiments support the interaction of NONO with TRF1 and TRF2 (Supplementary Figure 2C, D). Confocal microscopy did not reveal a significant

co-localization between NONO or SFPQ with PML, suggesting that the studied RNA binding proteins do not represent central components of APBs (Supplementary Figure 1E, F). Performing telomere ChIP we confirm association of SFPQ and NONO with telomere repeat chromatin and exclude binding to AluY

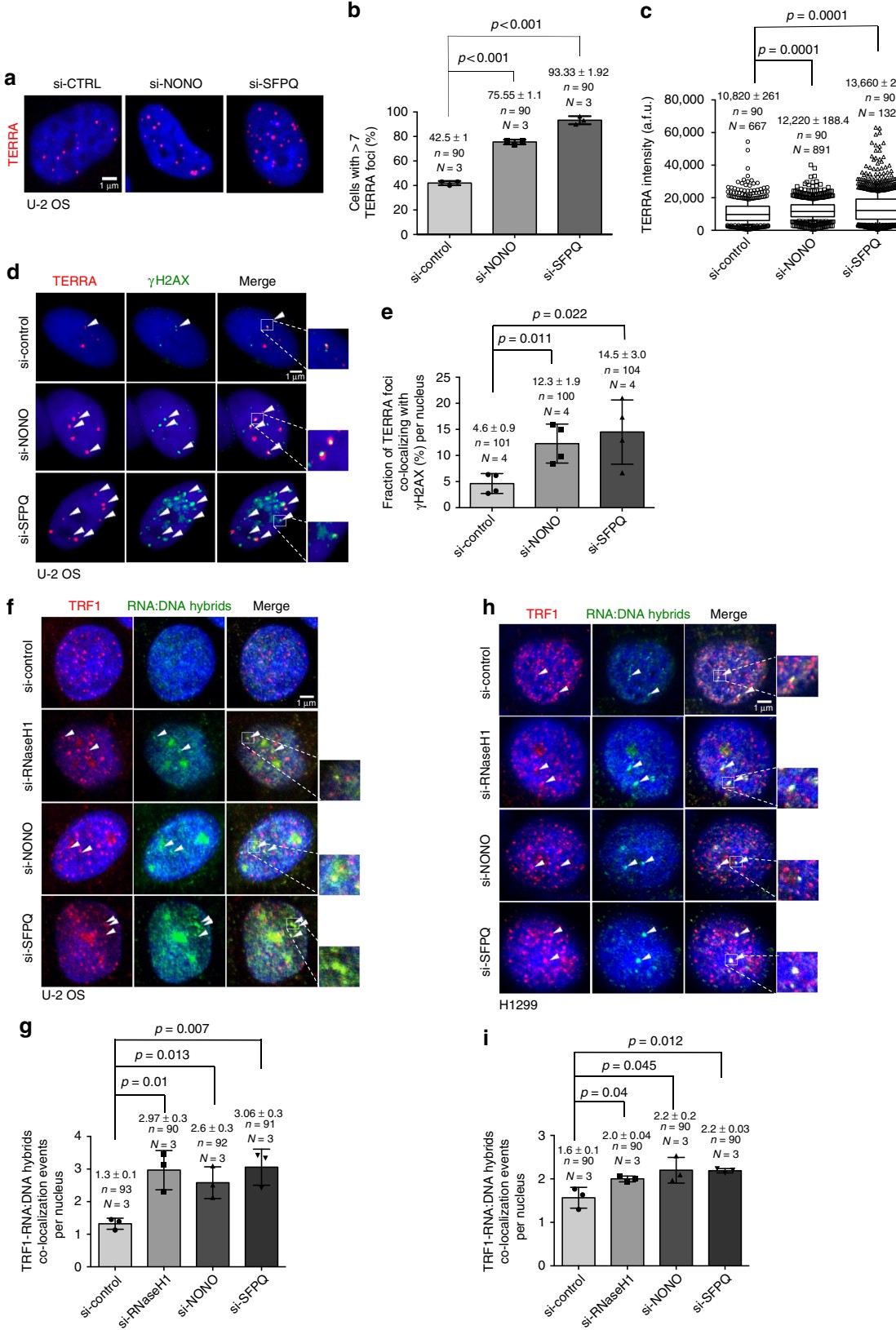

**Fig. 2** Depletion of NONO and SFPQ cause accumulation of RNA:DNA hybrids. **a** Representative images of TERRA RNA-FISH performed in U-2 OS cells transiently transfected with NONO- or SFPQ-specific siRNAs or with a non-targeting control siRNA. **b** Quantification of the percentage of cells with more than three TERRA foci per nucleus. **c** Quantification of TERRA signals intensity. Arbitrary fluorescence units (a.f.u.) are shown. Box plots: middle line represents median, and the box extends from the 25th to 75th percentiles. The whiskers mark the 10th and 90th percentiles. *p*-values were calculated using a two-tailed Mann–Whitney test. Median a.f.u. values and standard deviation are indicated; *n* = number of analyzed nuclei, *N* = number of analyzed TERRA signals. **d** Representative images of TERRA RNA-FISH combined with anti-γH2AX immunostaing performed in U-2 OS cells transfected with indicated siRNAs. Arrowheads indicate co-localization events. **e** Quantification of **d**; percentage of TERRA foci co-localizing with γH2AX per nucleus is indicated. **f, h** Representative confocal microscope images of combined immunofluorescence using S9.6 anti-RNA:DNA hybrid and anti-TRF1 antibodies in U-2 OS (**f**) or H1299 (**h**) cells transfected with indicated siRNAs. Arrowheads indicate co-localization events. **g, i** Quantification of telomeric RNA:DNA hybrids per nucleus in U-2 OS (**i**) or H1299 (**h**) cells. For quantifications in **b**, **e**, **g**, **i** mean values are indicated; error bars indicate standard deviation; *N* = number of independent experiments. *n* = number of analyzed nuclei. A two-tailed Student's *t*-test was used to calculate statistical significance; *p*-values are shown. Source data is provided as a Supplementary Information File. Scale bar, 1 μm. siRNAs listed in Supplementary Table 1

repeats (Fig. 1g, h). These data indicate that SFPQ and NONO represent TERRA interacting proteins that localize to telomere repeat sequences.

NONO and SFPQ cover multiple biological functions that include transcriptional regulation, the formation of subnuclear structures such as paraspeckles, but also DNA damage repair, gene expression, cell cycle control, and circadian rhythm[43,44]. In this study, we exclusively focus on the role of NONO and SFPQ in controlling telomere function in human cancer cells.

**NONO and SFPQ suppress telomere RNA:DNA hybrids and damage.** We next wished to understand whether NONO and SFPQ impact on TERRA expression in U-2 OS cells that produce TERRA at levels readily detectable by RT-PCR, Northern blotting and RNA-FISH. Northern blotting and quantitative RT-PCR using primers that amplify subtelomeric portions of TERRA originating from chromosomes 1 and 21 or chromosomes 2, 10, and 13 revealed that RNAi-mediated depletion of NONO or SFPQ does not impact on total TERRA levels (Supplementary Figure 2A-C; Supplementary Table 4). However, RNA-FISH revealed that knockdown of NONO or SFPQ in U-2 OS ALT cells significantly increased the number of TERRA foci per nucleus (Fig. 2a, b). In line with this, the proportion of cells with high TERRA foci number (>12) was increased in experimental cells (Supplementary Figure 2D). Finally, depletion of SFPQ resulted in a significant increase in TERRA RNA-FISH signal intensity (Fig. 2a–c). These data suggest that NONO and SFPQ do not impact on total TERRA levels but rather alter TERRA homeostasis at telomeres. Importantly, in NONO- or SFPQ-depleted interphase U-2 OS cells TERRA foci showed increased co-localization frequency with the DNA damage marker γH2AX (Fig. 2d, e). Analysis of metaphase chromosomes by immuno DNA-FISH showed increased localization of γH2AX at telomere sequences at chromosome ends in experimental H1299 and U-2 OS cells (Supplementary Figure 2E-H). Activation of a DNA damage response was validated by western blotting as shown by increased γH2AX levels and stabilization of p53 in U-2 OS cells (Supplementary Figure 2I, J). Together, this indicates that loss of NONO and SFPQ leads to altered TERRA homeostasis and promotes the formation of DNA damage at telomeres. Recent studies showed that RNA:DNA hybrid and R-loop formation at telomeres is linked with increased abundance of TERRA at telomeric chromatin, without altering total TERRA lncRNA levels[17,36]. To test whether NONO or SFPQ depletion enhances RNA:DNA-hybrid formation between TERRA and the telomeric C-strand, we performed co-immunocytochemistry using specific anti-TRF1 antibodies and affinity purified S9.6 monoclonal antibodies that detect RNA:DNA hybrids in a sequence independent manner. Confocal microscopy revealed that RNAi-mediated depletion of NONO or SFPQ significantly increased co-localization frequencies between RNA:DNA hybrid and TRF1

foci in both, U-2 OS and H1299 cells (Fig. 2f–i). This result was re-capitulated in RNAseH1 knockdown cells that have been recently reported to show increased abundance of telomeric RNA:DNA hybrids (Fig. 2f–i, Supplementary Figure 2K)[17]. RNAi-mediated depletion of TRF1 from U-2 OS cells resulted in a compete loss of immunostaining for TRF1, thus excluding cross reactivity of the anti-TRF1 antibody in our experimental setup (Supplementary Fig. 2L-M). Together, this indicates that NONO and SFPQ have a role in suppressing TERRA:telomere RNA:DNA-hybrid formation in telomerase-positive and negative cancer cells.

**NONO and SFPQ suppress replication defects at telomeres.** RNA:DNA hybrids trigger the formation of R-loop structures that represent obstacles during DNA replication, leading to replicative stress, DNA damage, and increased recombination frequencies at telomeres. Induction of replication defects at telomeres is linked with the phosphorylation of ATR and the phosphorylation of Serine 33 of the 32 kDa subunit of the Replication protein A (RPA32pSer33), both surrogate markers for replication stress[17]. As expected, induction of replicative stress by treatment of U-2 OS with hydroxyurea results in increased global RPA32pSer33 staining (Fig. 3c). In line with increased RNA:DNA-hybrid formation, we found that depletion of NONO or SFPQ from U-2 OS cells is paralleled by an increased co-localization of TERRA with phosphorylated RPA32 (Fig. 3a, b). This effect is paralleled by a significantly increased co-localization of TRF2 with RPA32pSer33 (Fig. 3c, d). Telomerase-positive H1299 cells recapitulate increased RPA32Ser33 phosphorylation at telomere in the absence of SFPQ. However, in loss of NONO conditions the increase in TRF2/RPA32Ser33 co-localization rates does not reach significance in H1299 cells (Supplementary Figure 3A, B). We propose that this is due to the increased resistance of telomerase-positive cancer cells to replication stress at telomeres, as previously shown for HeLa and H1299 cells[17,45]. Replicative stress is reported to trigger the phosphorylation of Ataxia telangiectasia and Rad3-related (ATR). In line with this we found an efficient phosphorylation of ATR at telomeres in SFPQ knockdown U-2 OS or H1299 cells (Fig. 3e, f; Supplementary Figure 3C, D). Remarkably, although loss of NONO triggers RPA32Ser33 phosphorylation at telomeres in U-2 OS cells, we were not able to observe increased telomeric ATR phosphorylation under these conditions. The role of NONO in enhancing ATR-mediated DNA damage response signaling may provide an explanation for this observation[43,46].

We next wished to test whether increased RNA:DNA-hybrid formation in NONO/SFPQ-depleted cells is directly linked to phosphorylation of RPA32Ser33. To address this issue, we aimed to rescue RPA32Ser33 phosphorylation levels in NONO- and SFPQ-depleted cells by ectopically expressing a mCherry-tagged version of human RNaseH1. As expected, co-depletion of NONO

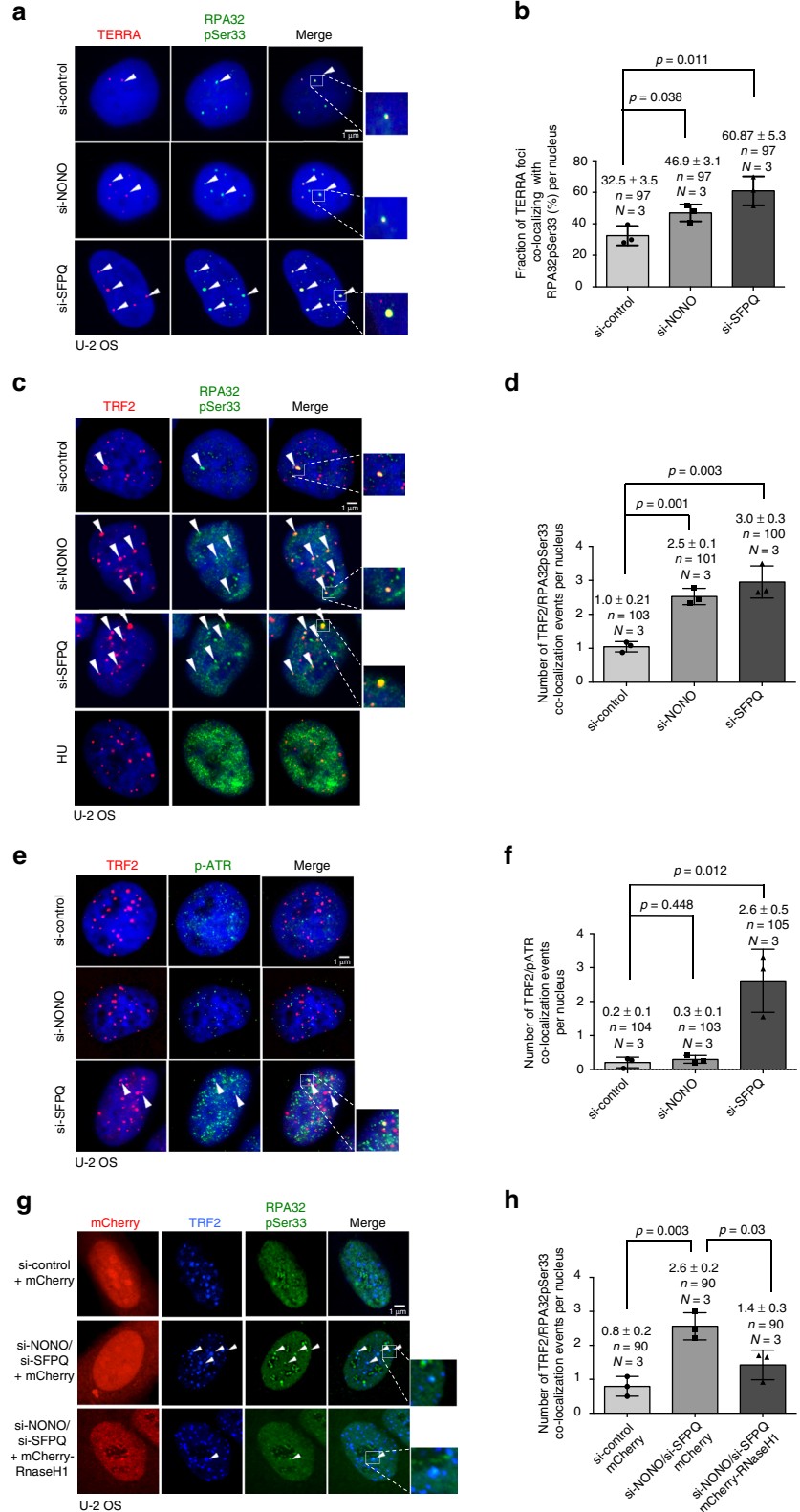

and SFPQ resulted in recruitment of RPA32Ser33 to telomeres of U-2 OS cells, indicative for replicative stress (Fig. 3g, h). Importantly, this effect is eliminated upon transient expression of RNaseH1 in NONO/SFPQ-depleted cells (Fig. 3g, h). Together, these data indicate that NONO and SFPQ are TERRA interacting proteins that have a crucial role in preventing RNA:DNA-hybrid

accumulation and R-loop-related replication defects at telomere repeat sequences.

**NONO and SFPQ regulate telomere leading strand fragility.** Increased RNA:DNA-hybrid levels at telomeres were shown to

**Fig. 3** Depletion of NONO and SFPQ cause replication defects at telomeres in U-2 OS cells. **a** Representative images of TERRA RNA-FISH combined with anti-RPA32pSer33 immunostaining performed in U-2 OS cells transfected with indicated siRNAs. Arrowheads indicate co-localization events. **b** Quantification of **a**; percentage of TERRA foci co-localizing with RPA32pSer33 per nucleus is indicated. **c** Representative images of combined immunofluorescence with anti-TRF2 and anti-RPA32pSer33 antibodies in U-2 OS cells transfected with indicated siRNAs. Arrowheads indicate co-localization events. Induction of replicative stress by treating cells with hydroxyurea (5 mM) for 6 h was used as control for RPA32pSer33 staining. **d** Quantification of TRF2/RPApSer33 co-localization events per nucleus. **e** Representative images of combined immunofluorescence using anti-TRF2 and anti-pATR antibodies (S428) in U-2 OS cells transfected with indicated siRNAs. Arrowheads indicate co-localization events. **f** Quantification of TRF2/pATR co-localization events per nucleus. **g** Representative images of combined immunofluorescence using anti-TRF2 and anti-RPA32pSer33 antibodies on U-2 OS cells transfected with indicated siRNAs and transiently expressing mCherry-RNase H1 or mCherry empty vector. Arrowheads indicate co-localization events. **h** Quantification of the number of TRF2/RPA32pSer33 co-localizations per nucleus. For quantifications in **b**, **d**, **f**, **h**, mean values and standard deviations are reported, error bars indicated standard deviation. $N$ = number of independent experiments. $n$ = number of analyzed nuclei. A two-tailed Student's $t$-test was used to calculate statistical significance; $p$-values are shown. Source data is provided as a Supplementary Information File. Scale bar, 1 μm. siRNAs listed in Supplementary Table 1

fuel R-loop formation and fragility of the telomeric leading (C-rich) strand in vertebrate cells[17,36]. Telomeric R-loop structures are prone to breakage, thus providing DNA substrates for homologous recombination at yeast and human telomeres. In order to understand the direct importance of NONO and SFPQ for telomere integrity, we performed short term loss of function experiments using telomere chromosome orientation DNA-FISH (CO-FISH). In this method BrdU incorporation during S-Phase and subsequent enzymatic digestion of the neo-synthesized DNA strand after UV treatment allows the selective detection of the TTAGGG containing lagging telomeric strand or the CCCTAA-repeat containing leading telomeric strand using differentially labeled, telomere-strand-specific DNA-FISH probes (Fig. 4a). This method allows to detect telomere-strand-specific aberrations but also telomere recombination events (Figs. 4a, 5a). RNAi-mediated depletion of NONO in U-2 OS cells significantly increased the appearance of aberrantly shaped or multi-dotted telomere signals the CCCTAA-repeat containing telomeric strand in U-2 OS cells (+60%; Fig. 4b). Interestingly, loss of NONO does not have an impact on telomere fragility at the TTAGGG repeat containing lagging strand (Fig. 4b). This data is in line with leading strand fragility triggered by increased RNA:DNA-hybrid abundance in RNaseH1 or Flap endonuclease loss of function cells[17,36]. As expected, ectopic expression of NONO reduced basal levels of telomere lagging strand fragility (Fig. 4c). Remarkably, loss of NONO in telomerase-positive cells results in telomeric leading and lagging strand fragility; accordingly, NONO over-expression reduces basic telomere fragility levels on both telomere strands (Supplementary Figure 4A, B). Together, these data identify NONO as novel suppressor of telomere fragility that has a particular relevance in suppressing fragility at the telomeric leading strand in ALT cells that are reported to be prone to exhibit telomeric RNA:DNA hybrids and leading strand fragility. NONO and SFPQ preferentially form heterodimers and have a common function in suppressing TERRA:telomere-hybrid formation. However, RNAi-mediated depletion of SFPQ had an advert effect on telomere fragility in U-2 OS and H1299 cells when compared to loss of NONO expression conditions. In fact, loss of SFPQ selectively reduces leading strand fragility in both, U-2 OS ALT and telomerase-positive H1299 cells (Fig. 4d; Supplementary Figure 4C). SFPQ overexpression does not impact on telomere fragility, suggesting that endogenous SFPQ expression levels are sufficient to keep telomere fragility at sustainable levels (Fig. 4e; Supplementary Figure 4D). Our data shows that SFPQ and NONO, although reported to preferentially act as heterodimer[43], have different impact on telomere fragility. In particular, NONO appears to be important to suppress telomere fragility; in contrast, loss of SFPQ appears to activate molecular processes that suppress telomere fragility.

**SFPQ and NONO suppress recombination in cancer cells**. To better understand the different function of NONO and SFPQ in the control of telomere stability we used telomere CO-FISH to study recombination-based telomere sister chromatid exchange (T-SCE), a reported downstream result of RNA:DNA-hybrid-related R-loop formation at eukaryotic telomeres (Fig. 5a)[15,17]. We found that NONO depletion does not significantly impact on T-SCE in U-2 OS and H1299 cancer cells (Fig. 5b, c). In contrast, SFPQ knockdown results in a significant increase in T-SCE frequency in both, U-2 OS ALT and telomerase-positive H1299 cells (Fig. 5b, c). This identifies SFPQ as a novel repressor of telomere recombination in both, telomerase-positive and telomerase-negative human cancer cells. Activation of ATR was recently shown to promote homologous recombination, providing an explanation for ATR phosphorylation in SFPQ, but not NONO knockdown cells (Fig. 3e, f; Supplementary Figure 3C, D)[47]. We next wished to better understand a potential contribution of telomere fragility in NONO-depleted cells to the regulation of T-SCE events. We found that in the context of increased telomere fragility caused by NONO knockdown, co-depletion of SFPQ resulted in a dramatic 18-fold increase of T-SCE that involves 35% of detectable telomeres (Fig. 5d, e). This effect was paralleled by a rescue of telomere fragility in NONO/SFPQ double knock-down cells (Fig. 5d–f). This suggests that telomere sister chromatid recombination may represent a mechanism to resolve telomere fragility triggered by NONO depletion. This mechanism can also explain reduced basal levels of telomeric leading strand fragility in SFPQ-depleted H1299 and U-2 OS cells (Fig. 4d; Supplementary Fig. 4C). In line with increased T-SCE frequency in NONO/SFPQ-depleted cells we observed an increased frequency of co-localization of TRF2 with Promyelocytic Leukemia (PML) nuclear bodies and co-localization of RAD51 with TRF2 in U-2 OS cells that were depleted for SFPQ (Fig. 5g–j). Accordingly, we found significantly elevated TERRA–PML co-localization in SFPQ knockdown cells, a feature reported for ALT cells (ref. [17] Supplementary Figure 5A, B). This is indicative for an enhanced generation of alternative lengthening of telomeres-associated PML nuclear bodies (APBs) and increased homologous recombination at telomeres under SFPQ loss of function conditions. As expected, depletion of NONO does not impact on APB frequency in U-2 OS cells. Remarkably, although siRNA-mediated depletion of SFPQ in telomerase-positive H1299 cells triggers T-SCE we did not observe significantly increased APB numbers (Fig. 6e, f). This suggests that loss of SFPQ can trigger telomere recombination in telomerase-positive cancer cells; however, without efficiently engaging APBs.

Together, our data indicate that NONO and SFPQ have distinct roles in ensuring telomere stability. NONO suppresses the generation of R-loop-related leading telomere fragility; in

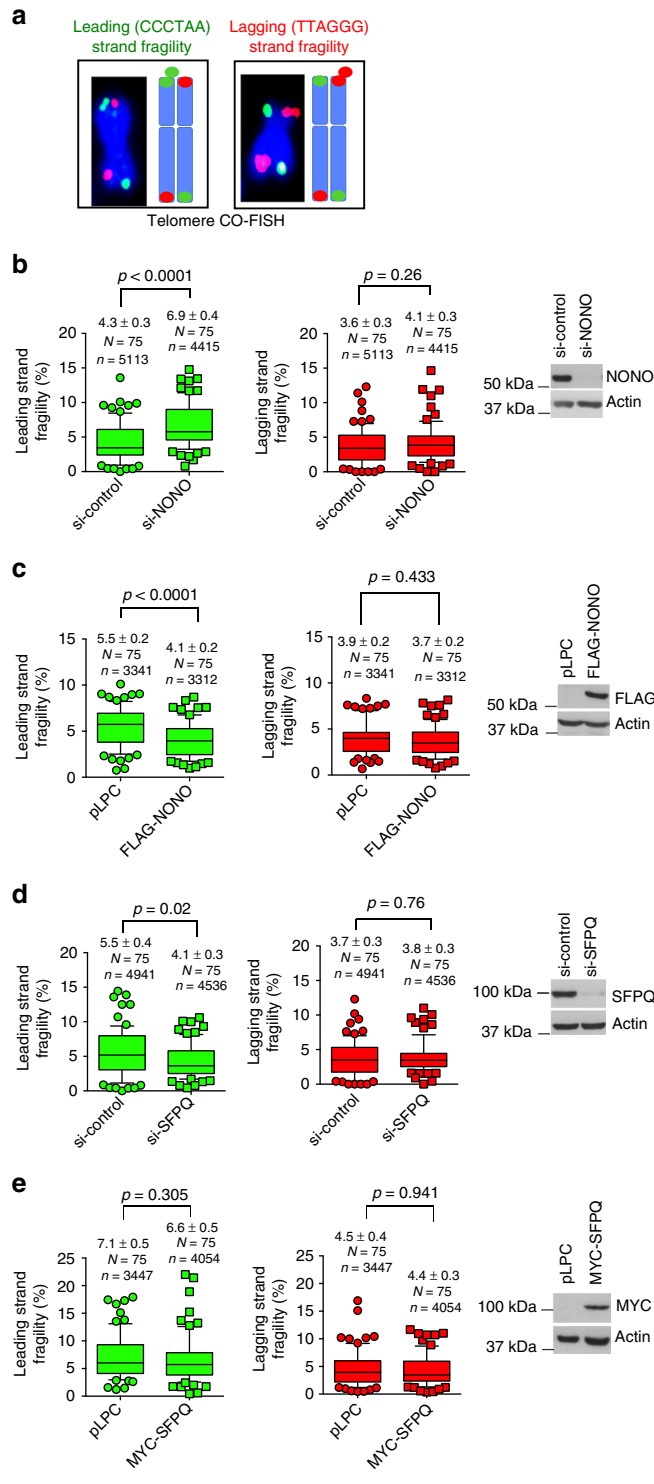

**Fig. 4** NONO and SFPQ regulate telomeric leading strand fragility of U-2 OS cells. **a** Representative images and explanative models of leading strand and lagging strand telomeric fragility as detected by Chromosome Orientation FISH (CO-FISH). Fragile telomeres appear as multiple telomere signals. **b** Quantification of leading strand fragility (left panel) and lagging strand fragility (central panel) in U-2 OS cells transfected with NONO-specific siRNA or a non-targeting control siRNA. Right panel, western blotting showing NONO knockdown efficiency. **c** Quantification of leading strand fragility (left panel) and lagging strand fragility (central panel) in U-2 OS cells that stably overexpress FLAG-NONO or a control vector. Right panel, western blotting showing ectopic expression of FLAG-tagged NONO. **d** Quantification of leading strand fragility (left panel) and lagging strand fragility (central panel) in U-2 OS cells transfected with SFPQ-specific siRNA or a non-targeting control siRNA. Right panel, western blotting showing SFPQ knockdown efficiency. **e** Quantification of leading strand fragility (left panel) and lagging strand fragility (central panel) in U-2 OS cells that stably overexpress myc-tagged SFPQ or an empty vector. Right panel, western blotting showing expression of ectopic myc-SFPQ. For quantifications in **b**–**e**, data points represent the percentage of fragile telomeres per metaphase spread. Only chromosome ends with detectable telomeres were considered for analysis. Metaphases from three independent experiments were analyzed. Box plot diagrams (**b**–**e**): middle line represents median; boxes extend from the 25th to 75th percentiles. The whiskers mark the 10th and 90th percentiles. *p*-values were calculated using a two-tailed Mann–Whitney test. Median fragility values and standard deviation are indicated; *n* = number of analyzed telomere repeat signals, *N* = number of analyzed metaphase spreads. siRNAs listed in Supplementary Table 1. Source blots are available in Supplementary Figure 8

**SFPQ and NONO control telomere length in cancer cells.** Recombination-based ALT ensures telomere maintenance in telomerase negative cancer cells[37,38]. Accordingly, the impairment of pathways related to homologous recombination results in alterations in telomere length homeostasis in ALT cells[48,49]. Therefore, we were interested to study the role of SFPQ and NONO in telomere length homeostasis in telomerase negative U-2 OS ALT cells but also telomerase-positive H1299 cells. Quantitative telomere DNA-FISH revealed that siRNA mediated, transient depletion of NONO or SFPQ in U-2 OS ALT cells for 3 days induces a 8% or 17% increase in telomere length, respectively (Fig. 6a, b). This telomere elongation phenotype is exacerbated upon combined depletion of NONO and SFPQ (+ 35%) (Fig. 6a, b). This is in line with the dramatically increased T-SCE frequency under these conditions (Fig. 5d, e). In contrast to telomere elongation in U-2 OS cells, we found that unleashing telomere recombination by single depletion of SFPQ or combined depletion of NONO and SFPQ resulted a 21% reduction of telomere length in telomerase-positive H1299 cells, respectively (Fig. 6c, d). We thus hypothesize that enhancing T-SCE rates in absence of functional APBs leads to an inefficient homologous recombination process at telomeres in SFPQ knockdown H1299 cells, resulting a net loss of telomeric repeats (Fig. 6c, d). To test this hypothesis, we aimed to rescue the telomere shortening phenotype by stimulating the efficiency of homologous recombination pathways in SFPQ-depleted H1299 cells. SFPQ was depleted from telomerase-positive H1299 cells that were co-treated with 5-Aza-2′-deoxycytidine to unleash recombination-based telomere elongation. 5-Aza-2′-deoxycytidine treatment has been reported to reduce subtelomeric DNA methylation, thus driving recombination events in cancer cells, that are detectable by telomere CO-FISH[50]. As expected, 5-Aza-2′-deoxycytidine treatment lead to increased APB frequency in H1299 cells

contrast, SFPQ has a critical role in suppressing uncontrolled recombination events triggered by the accumulation of telomere R-loops and replication stress.

We further propose that increased telomere fragility in NONO-depleted cells increases the availability of recombinogenic substrates at telomeres that efficiently engage in homologous recombination in the absence of SFPQ. We thus propose that NONO and SFPQ collaborate to suppress RNA:DNA-hybrid triggered telomere recombination in both ALT- and telomerase-positive cancer cells.

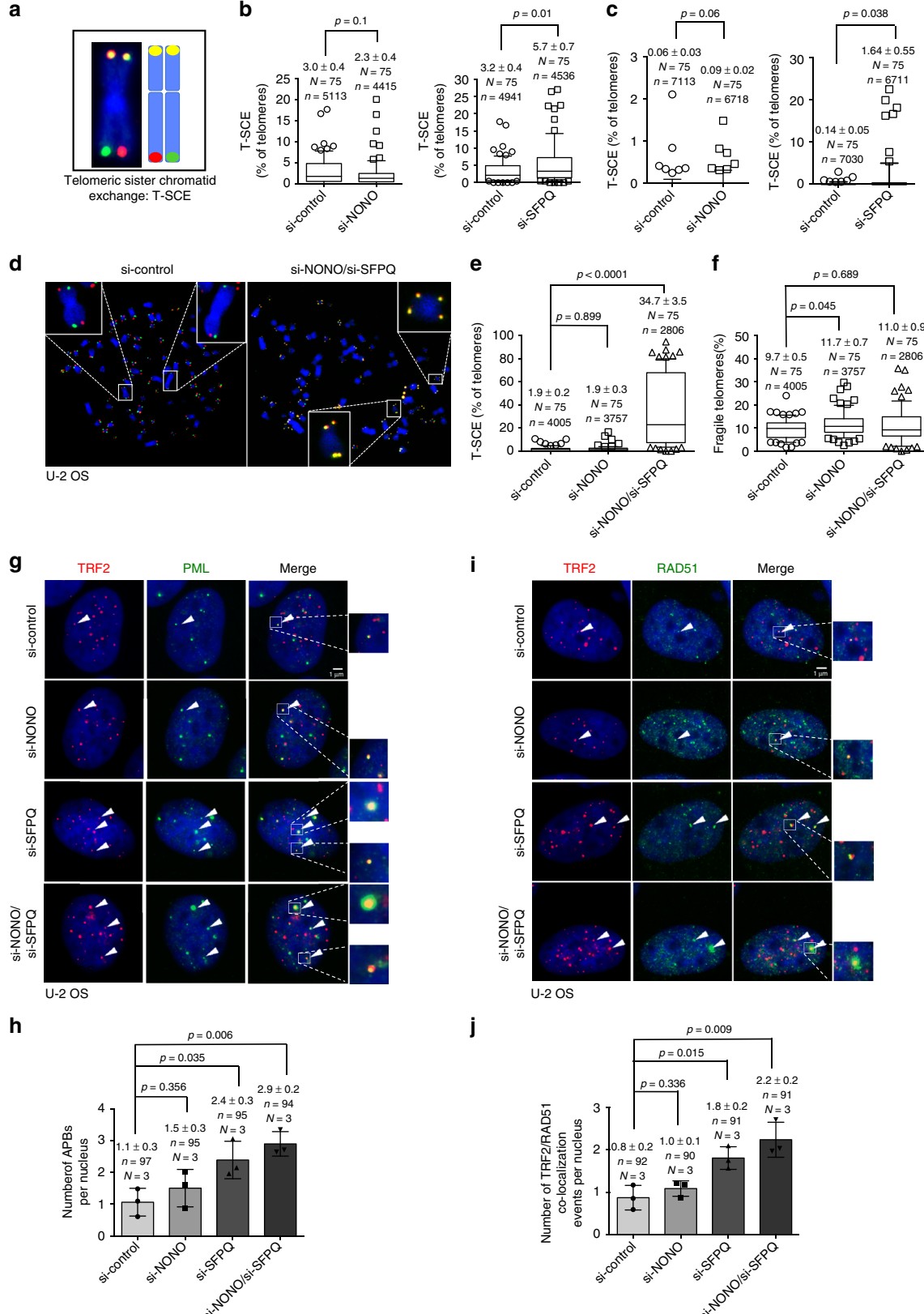

(Fig. 6e, f). This effect was found to be enhanced in SFPQ-depleted and 5-Aza-2′-deoxycytidine-treated H1299 cells, suggesting the activation of a functionally relevant subnuclear compartment for the ALT pathway (Fig. 6e, f). Importantly,

under these conditions, quantitative telomere DNA-FISH experiments revealed a reversion of the telomere shortening phenotype of SFPQ-depleted H1299 cells, as demonstrated by a 47% telomere elongation in 5-Aza-2′-deoxycytidine-treated SFPQ

**Fig. 5** NONO collaborates with SFPQ to suppress recombination at telomeres. **a** Representative images and explanatory models of telomeric sister chromatid exchange (T-SCE) as detected by CO-FISH. **b** Quantification of T-SCEs in U-2 OS cells transfected with control siRNAs, NONO-specific siRNAs (left panel), or SFPQ-specific siRNAs (right panel). **c** Quantification of T-SCEs in H1299 cells transfected with control siRNAs, NONO-specific siRNAs (left panel), or SFPQ-specific siRNAs (right panel). **d** Representative images of CO-FISH experiments performed in U-2 OS cells transfected with indicated siRNAs. **e** Quantification of T-SCEs in U-2 OS cells transfected with NONO-specific siRNA, or with both NONO- and SFPQ-specific siRNAs compared to control cells. **f** Quantification of telomere fragility in U-2 OS cells transfected with NONO-specific siRNA, or with both NONO and SFPQ-specific siRNAs. **g** Representative images of anti-TRF2 and anti-PML co-immunofluorescence to detect APBs in U-2 OS cells transfected with indicated siRNAs. **h** Quantification of the number of APBs per nucleus of data shown in **g**. **i** Representative images of anti-TRF2 and anti-RAD51 co-immunofluorescence in U-2 OS cells transfected with indicated siRNAs. **j** Quantification of RAD51-TRF2 co-localization events per nucleus of data shown in **i**. Box plot diagrams (**b**, **c**, **e**, **f**): middle line represents median; boxes extend from the 25th to 75th percentiles. The whiskers mark the 10th and 90th percentiles. *p*-values were calculated using a two-tailed Mann–Whitney test. Median T-SCE/fragility values and standard deviation are indicated; n = number of analyzed telomere repeat signals, N = number of analyzed metaphase spreads. Panels **h**, **j** means (bars) and standard deviation (error bars) are reported. N = number of independent experiments. n = number of analyzed nuclei. A two-tailed Student's *t*-test was used to calculate statistical significance; *p*-values are shown. Source data is provided as a Supplementary Information File. Scale bar, 1 μm. siRNAs listed in Supplementary Table 1

knockdown H1299 cells, when compared to control cells (Fig. 6g, h). This indicates that functional subnuclear APB compartments are necessary to convert enhanced telomere recombination of SFPQ-depleted telomerase-positive cells into a telomere elongation phenotype. We thus conclude that NONO and SFPQ are novel regulators of telomere length in both telomerase-positive and negative cancer cells.

Here, we show that RNA binding proteins NONO and SFPQ are novel component of telomeric chromatin that suppress replicative stress at telomeres by antagonizing the formation of RNA:DNA hybrids. Remarkably, although preferentially acting in heterodimers, NONO and SFPQ control different pathways that result from the formation of RNA:DNA hybrids and R-loop structures: NONO appears to suppress telomere fragility; in contrast, SFPQ has a central role in suppressing homologous recombination (Fig. 7). Combined loss of NONO and SFPQ depletion results in a massive increase in homologous recombination and altered telomere length homeostasis. Altogether, our data highlight a role for NONO and SFPQ in controlling RNA: DNA-hybrid-related telomere instability and telomere length homeostasis in both telomerase-positive and negative cancer cells.

## Discussion
Here, we show that NONO and SFPQ are novel telomere repeat associated proteins that collaborate to suppress RNA:DNA-hybrid-related replicative stress, telomere fragility and telomere recombination in both U-2 OS ALT cells and H1299 telomerase-positive human cancer cells. Loss of NONO or SFPQ results in an increased appearance of TERRA foci without affecting total TERRA expression levels suggesting that trapping TERRA in RNA:DNA-hybrid structures may increase the amount of detectable TERRA transcripts at telomeres. In ALT cells, extra-chromosomal telomeric repeats (ECTR) can constitute a notable fraction of the total telomere repeat content, suggesting that a fraction of NONO and SFPQ may also locate to this type of telomere repeat sequences[51]. However, alterations at telomeres of metaphase chromosomes observed in loss of function experiments support a role for NONO and SFPQ at telomeres. We directly show that loss of NONO and SFPQ mediates enhanced formation of telomeric RNA:DNA hybrids in both, telomerase-positive and negative human cancer cells. A common role of NONO and SFPQ in suppressing RNA:DNA-hybrid abundance and replication defects in human cancer cells is in accordance with several studies that propose that NONO and SFPQ act on various aspects of RNA metabolism by preferentially forming heterodimers[43]. In line with the increased abundance of RNA:DNA hybrids, we observed increased appearance of the classic DNA damage marker γH2AX

at telomere repeats of NONO- or SFPQ-depleted cells. This is in line with independent studies that show an involvement of NONO and SFPQ in promoting DNA damage pathways, including homologous recombination, non-homologous end joining, nucleotide excision repair, and interstrand cross-link repair[43,44]. The fact that NONO and SFPQ contain multiple RNA binding domains suggests an "RNA-based" role of NONO and SFPQ in preventing RNA:DNA-hybrid-related DNA damage. Remarkably, although acting as heterodimers, NONO and SFPQ have apparently different roles in suppressing downstream events related to RNA:DNA hybrids at telomeres. NONO appears to play an important role in preventing fragility of the leading, CCCTAA-repeat containing telomeric strand. In line with this, leading strand fragility was previously shown to be a direct consequence of telomeric RNA:DNA hybrids in different model systems[17,36,52]. In contrast, CO-FISH analysis revealed that SFPQ acts as barrier to homologous recombination at telomeres. Loss of SFPQ drives RNA:DNA-hybrid formation, impaired replication and results in dramatically increased recombination frequencies in the context of ATR phosphorylation at telomeres of telomerase-positive and ALT cancer cells. This effect is paralleled by a reduction of telomere fragility, leading to the interesting speculation that homologous recombination may help to rescue telomere fragility. Our data show that SFPQ functions as barrier to homologous recombination at telomeres. Phosphorylation of ATR in SFPQ knockdown cells is in line with a recent report that demonstrated that activation of ATR promotes homologous recombination[47]. Remarkably, removing the SFPQ dependent recombination barrier in the context of increased fragility mediated by NONO depletion results in massive T-SCE events that involve 35% telomeres detectable by DNA-FISH in U-2 OS ALT cells. We propose that increased chromosome fragility in the absence of NONO leads to the generation of elevated numbers of recombinogenic DNA products that efficiently engage in telomere recombination when removing the recombination barrier SFPQ. Thus, NONO and SFPQ have a direct role in suppressing RNA:DNA-hybrid-related telomere fragility and recombination (Fig. 7). Given the fact that NONO and SFPQ do not contain enzymatic activity and may rather function as scaffold we propose that localization of NONO or SFPQ to telomeres is necessary to recruit specific—yet to define—factors that suppress RNA:DNA-hybrid-related telomere fragility and recombination[43,44]. Homologous recombination represents a key pathway for the regulation of telomere length homeostasis. In line with this, we found that NONO/SFPQ knockdown in U-2 OS triggers increased T-SCE frequencies, increased localization of RAD51 at telomere repeats as well as the engagement of telomeres in APBs, thus resulting telomere elongation. Under the same conditions, telomerase-positive H1299

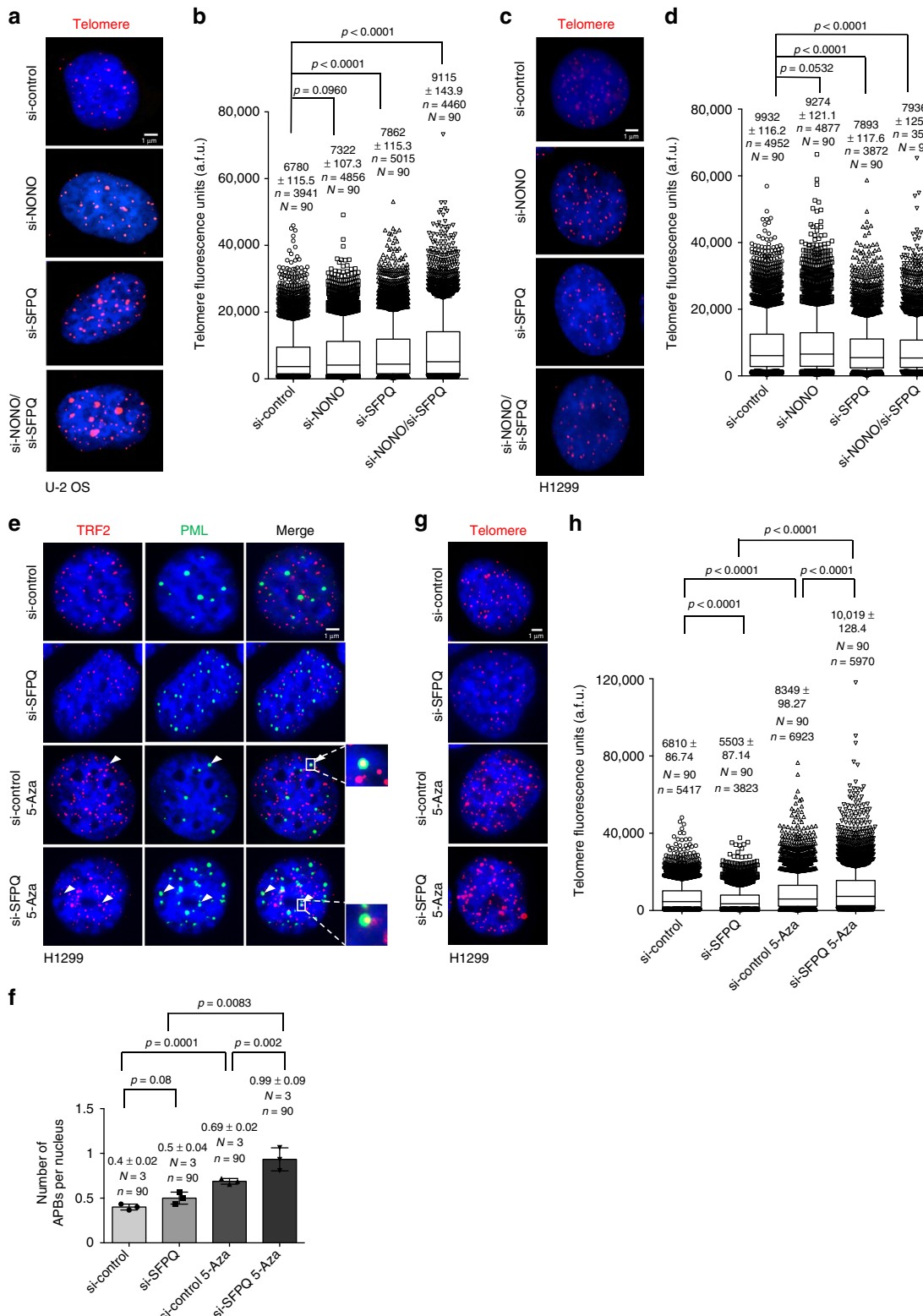

cells, that lack functional APB compartments, show rapid telomere shortening. Remarkably, telomere shortening can be rescued by 5-Aza-2′deoxycytidine treatment that forces APB formation and T-SCE in SFPQ loss of function cells[50]. This suggests that functional APBs are important to convert increased T-SCE frequencies in SFPQ loss of function cells into an overall increase in telomere length.

Together, this identifies NONO and SFPQ as regulators of telomere length homeostasis by controlling RNA:DNA-hybrid abundance at telomeres. Given the relevance of RNA:DNA hybrids in recombination and telomere length homeostasis, NONO and SFPQ are expected to be of special relevance in human tumors that maintain telomeres length via the ALT pathway. In line with this, SFPQ mutations have been recently

**Fig. 6** NONO and SFPQ regulate telomere length homeostasis in human cancer cells. **a** Telomere length measurements by quantitative telomere DNA-FISH on interphase U-2 OS cells after transient transfection with indicated siRNAs. Representative images are shown. **b** Telomere fluorescence intensity was analyzed for each telomere of **a**. **c** Telomere length measurements by quantitative telomere DNA-FISH on interphase H1299 cells after transient transfection with indicated siRNAs. Representative images are shown. **d** Telomere fluorescence intensity analyzed for each telomere of **c**. **e** Representative images of combined immunofluorescence using anti-TRF2 and anti-PML antibodies in H1299 cells transfected with SFPQ-specific siRNA or a non-targeting control siRNA. Cells were treated or not treated with 5-aza-2′-deoxycytidine (5-Aza). **f** Quantification of **e**; number of TRF2-PML co-localization events per nucleus. Mean values (bars) and standard deviations (error bars) are reported. N = number of independent experiments. n = number of analyzed nuclei. A two-tailed Student's t-test was used to calculate statistical significance; p-values are shown. Source data is provided as Supplementary Information File. **g** Representative images of telomere length measurements by quantitative telomere DNA-FISH on interphase H1299 cells transfected with SFPQ-specific siRNA or a non-targeting control siRNA. Cells were treated or not treated with 5-aza-2′-deoxycytidine (5-Aza). **h** Telomere fluorescence intensity analyzed for each telomere of **g**. For telomere length measurements, Box plot diagrams (**b**, **d**, **h**): middle line represents median arbitrary fluorescence units (a.f.u.), boxes extend from the 25th to 75th percentiles. The whiskers mark the 10th and 90th percentiles. p-values were calculated using a two-tailed Mann–Whitney test. Median a.f.u. values with standard deviation are indicated; n = number of analyzed telomere repeat signals, N = number of analyzed interphase nuclei. Arbitrary fluorescence units (a.f.u.) are shown. Scale bar, 1 μm. siRNAs listed in Supplementary Table 1

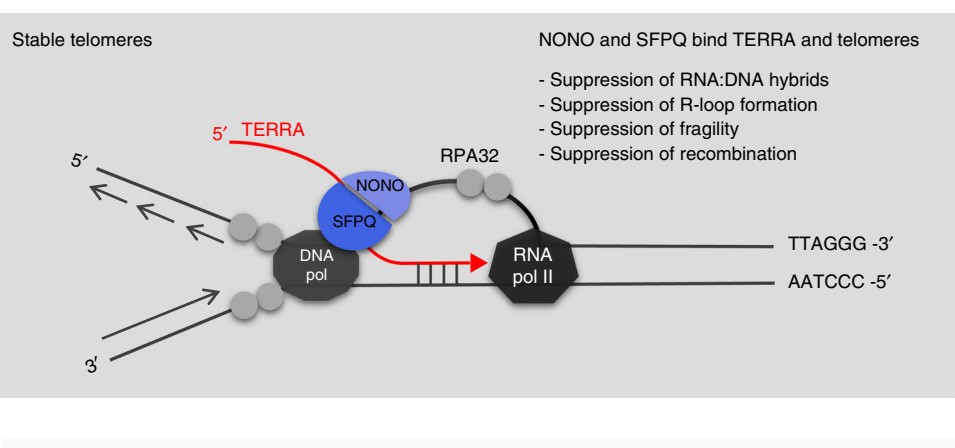

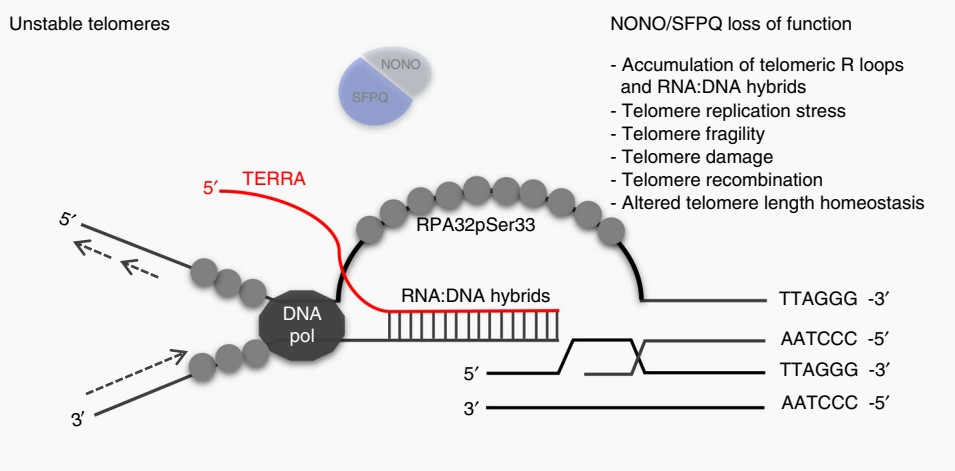

**Fig. 7** A model for the role of NONO and SFPQ in controlling telomere stability. Top panel: NONO and SFPQ bind TERRA transcripts at telomeres and control mechanisms that antagonize the accumulation of RNA:DNA hybrids, thereby stabilizing telomeres. Bottom panel: Loss of NONO and SFPQ leads to the accumulation of RNA:DNA hybrids at telomeres, resulting in the displacement of the non-template telomere strand (G-strand) and R-loop formation. Increased R-loop formation triggers replication defects, fragility, telomere damage and the exposition of recombinogenic telomeric DNA that engage in telomere recombination events and result alterations in telomere length homeostasis. DNA pol, DNA polymerase; RNA pol II, RNA polymerase II

discovered by exome sequencing in human osteosarcoma, a tumor type that is reported to use recombination-based ALT as predominant pathway for telomere maintenance[53]. We hypothesize that NONO and SFPQ act as platforms to recruit factors that resolve DNA:RNA-hybrid structures thereby suppressing fragility and recombination at telomeres (Fig. 7). Understanding the function of the entire NONO/SFPQ complex at telomeres is expected to open new avenues in understanding the suppression of the ALT pathway in telomerase-negative tumors and activation of the ALT pathway in telomerase negative cells. These insights are expected to be highly relevant for potential telomere-based anti-cancer therapies.

## Methods

**Immunofluorescence.** Cells were fixed in 4% paraformaldehyde (PFA) for 15 min, followed by treatment with citrate buffer [0.1% (w/v), 0.5% Triton X-100] for 5 min at room temperature. Cells were blocked for 45 min in 3% BSA, 0.1% Tween-20 in 1x PBS and incubated with primary antibodies (as indicated in Supplementary Table 2) diluted in blocking solution at room temperature for 2 h. Cells were washed in 0.3% BSA, 0.1% Tween-20 in 1x PBS and incubated with secondary antibodies (Supplementary Table 3) for one hour at room temperature. Slides were washed in 0.1% Tween-20 1XPBS, stained with DAPI (Sigma) and mounted in Vectashield (Vector laboratories). For S9.6 immunofluorescence, cells were fixed and permeabilized with ice-cold methanol for 10 min and acetone for 1 min on ice as previously described[23]. Blocking, antibody and washing solutions were performed in 4x SSC. Cells with at least 5 nuclear extra-nucleolar S9.6 foci were analyzed. Co-localization events were quantified using ImageJ 1.46r or by visual inspection. The Student's t-test was used to calculate statistical significance.

**RNA FISH.** Cells were permeabilized with three consecutive steps (30 sec each) of cytobuffer (100 mM NaCl, 300 mM Sucrose, 3 mM MgCl$_2$, 10 mM Pipes pH 6.8), cytobuffer/0.5% Triton-X, and cytobuffer. Cells were fixed for 10 min at room temperature in ice-cold 4% paraformaldehyde (PFA) in 1x PBS and then dehydrated twice in ice-cold 70% ethanol for 2 min, ice-cold 90% ethanol for 1 min, and ice-cold 100% ethanol for 1 min. Dried slides were incubated overnight with a Cy3-labeled TERRA probe in a humid chamber (2x SSC, 50% formamide) at 37 °C. Slides were washed once in 2x SSC for 3 min at room temperature and three times in 2x SSC for 5 min at 37 °C. Subsequently, slides were transferred to 4x SSC and mounted in Vectashield (Vector laboratories). The TERRA probe was labeled with the FISH Tag DNA kit (Invitrogen).

**Immunofluorescence combined with RNA FISH.** Cell were permeabilized with cytobuffer as described for RNA-FISH and then subjected to the immuno-fluorescence protocol. Used primary and secondary antibodies are indicated in Supplementary Table 2, 3. After the washing step for the secondary antibody, cells were fixed at room temperature for 2 min with 4% paraformaldehyde (PFA), dehydrated and RNA-FISH was performed. Co-localization of foci from FISH and immunostainings were quantified by visual inspection.

**Telomere DNA-FISH.** DNA-FISH was carried out as previously described[54]. Cells were fixed with 4% paraformaldehyde for 10 min at RT. After three washes with 1x PBS for 5 min, slides were dehydrated by washing in 70%, 90%, and 100% ethanol for 5 min each. Slides were allowed to dry for 10 min at RT. A Cy3-labeled (CCCTAA)$_3$ probe was added to sample and after denaturation at 80 °C for 3 min, the slides were incubated for 2 h at room temperature in a humid chamber. Subsequently, the slides were washed under agitation, twice with FISH solution (70% formamide, 10 mM Tris pH 7.2, 0.1% BSA) for 15 min at room temperature and three times with 0.01% Tween-20 in 1x TBS at room temperature. Slides were then dehydrated with washes in 70%, 90%, and 100% ethanol. Samples where mounted with Vectashield (Vector laboratories). Interphase nuclei were analyzed using spot IOD analysis (TFL-TELO) software. A Mann–Whitney was used to calculate statistical significance.

**Chromosome prientation FISH (CO-FISH).** Confluent experimental cells were subcultured in the presence of 5′-bromo-2′-deoxyuridine (BrdU, 10 μM; Sigma) and cultivated for 18–20 h. Colcemid was added during the final period of BrdU treatment (U-2 OS, 0.2 μg/m 3 h; H1299, 1 μg/ml for 2 h). Cells were recovered and metaphases prepared as previously described[54]. CO-FISH was performed as previously described[55,56] using first a (CCCTAA)$_3$ probe labeled with Cy3 and then a second (TTAGGG)$_3$ probe labeled with FITC. Samples were processed as described for DNA-FISH.

**Reporting summary.** Further information on experimental design is available in the Nature Research Reporting Summary linked to this article.

## Data availability

The data that support the findings of this study are available from the corresponding author upon reasonable request. Source data for western blot and ChIP experiments in Figs. 1 and 4 are available in the Supplementary Information File.

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

## Acknowledgements

The laboratory of S.S. is supported by the Italian Association for Cancer Research (AIRC) Start up grant 10299, the AIRC-IG 18381 grant, a FRA 2015 intramural grant from the University of Trieste, Italy, and a grant by the European Union, European Regional Development Fund, and Interreg V-A Italia-Austria 2014–2020 (ITAT1096-P). R.B is supported by a AIRC grant Rif 42/08, 6352 and 17756. E.P. was supported by an AIRC fellowship. We thank Alessandro Marcello and Fabrizio D'Adda di Fagagna for providing vectors and Maria Blasco for DNA-FISH probes.

## Author contributions

E.P., V.B., R.B., and S.S. designed experiments; E.P., V.B., A.Z., O.S., P.V.B., and R.D. carried out experiments; S.M. performed mass spectrometry analysis; E.P., V.B., R.B., and S.S. wrote the manuscript.

## Additional information

**Competing interests:** The authors declare no competing interests.

