## [Peer Review File · Nature Communications]

Reviewers' comments:

Reviewer #2 (Remarks to the Author):

In this manuscript Petti et al identify RNA interacting factors NONO and SFPQ as proteins that bind TERRA and interact with the telomere chromatin, and investigate putative roles for NONO and SFPQ in telomere biology through their TERRA associated roles. They conclude SFPQ and NONO help maintain telomere integrity through the modulation of R-loops to suppress telomere fragility and mediated homologous recombination. They also attempt to tie this activity to ALT-mediated telomere length maintenance. These are novel experiments that could influence thinking in the field, specifically regarding how RNA regulatory pathways impact telomere integrity and telomere length maintenance. While the novelty is appreciated, and their efforts are well-meaning, the manuscript unfortunately suffers from experimental and conceptual shortcomings that preclude publication. Areas of improvement are categorized into minor and major concerns, and where possible collated thematically.

Major thematic comments

1. **Sample size.** In general, for any experiment where nuclei are being analyzed, this requires a minimum of 3 experiments of 30 nuclei per experiment, for a minimal total of 90 nuclei. For metaphase analysis, a minimum of 3 experiments of 25 chromosome spreads per experiment for 75 metaphase spreads total. This impacts most of the data sets in the paper, which are typically underpowered.
2. **Statistics.** My understanding is that student t-tests determine if the means of two groups are statistically different, but this relies on normally distributed data. Many of the data shown here do not appear to be normal distributions (specifically in figure 6), therefore, a t-test would be invalid. Wilcoxon-Mann-Whitney rank-sum testing is likely a more appropriate statistical test, which will determine if there are differences in the data distribution. As you are measuring larger datasets (i.e. distribution in the length of many telomeres), it is more appropriate to investigate differences in distribution as opposed to means. The data sets in question include 2A, 5B, 5C, 5E, 5F, 6A, 6B, 6E, and potentially the data in figure 4. Many of these will remain statistically significant. However, I have concerns for the quantitative data in figure 6 which does not appear to be significant.
3. **Data presentation.** Bar graphs are better represented by showing individual symbols for the mean of each replicate, with the experimental mean as a line, and error bars showing standard deviation or error. Replicate variability is hidden in the bar graph format. This is appropriate for small data sets ($n = 3$ experiments). For larger data sets with many data points (i.e. > 50 data points) I suggest a box and whiskers graph. For example, the quantitative data in figure 2C, and 6A, B, E are hard to decipher at dot plots given the density of data points, and would be better represented with box and whiskers.
4. **Image quality and representation.** [A] Most of the data in this manuscript rely heavily on imaging data. There are differing problems with the data sets. Some images appear to be heavily saturated, making it difficult to interpret the outcomes (e.g. 2F, 2H, 5G, 6A, Supplementary 2H, 2I, 3A). [B] In other data sets, co-localizations are scored, but one of the channels is diffuse puncta. For example, in 1D, 1E, it is hard to determine if the telomeres co-localize specifically with NONO/SFPQ or if this is a result of extensive NONO/SFPQ staining throughout the nucleus. Specific co-localization can be determined by shifting one channel several pixels, and re-counting. If the co-localizations remain similar in number after the pixel shift, it indicates these are not specific co-localizations, but simply random overlap of abundant puncta. Alternatively, for this experiment, cytocentrifugation as shown in Supplementary Figure 2D but probed with anti-NONO or SFPQ might work well to show telomere specific localization. Similar problems arise with Figure 3A and 3C. [C] There appears to be bleed through in the RNA:DNA hybrid staining to the telomere channel – as shown by retained TRF1 signal in Supplementary 2I following TRF1 knockdown. This brings the data on RNA:DNA hybrid imaging into question. [D] I do not see a lagging fragile telomere in the example in 4A. The red telomere, which I infer from the diagram is supposed to be fragile,

look to me to be an over-saturated telomere signal. This raises questions about data scoring of these experiments. [E] For all data sets, it is unclear if presented images are single focal planes, maximum intensity projects, etc? This is not described but are important information. [F] The use of arrows to identify foci is nice, but in some images the arrows are so numerous it obscures the image. It is better to show one or two examples and let the reader infer the remainder from the image.

Major experimental comments

1. How many times was the ChIP experiment done in figure 1, and where is the quantitation? Additionally, it is appropriate to use anti-H3 as a positive control for both telomere and Alu. Currently there is no indication of the researchers ability to capability to ChIP Alu repeats, limiting its utility as a non-telomere control.
2. For images in 2A, 2D, 2H and all of figure 3 – are you looking at telomeres or APBs? The number of TERRA or TRF2 foci are too few to be telomeres, additionally the foci are often large and bright, consistent with APBs. These experiments were done in ALT cells. Does this indicate NONO and SFPQ act at APBs instead of chromosomal telomeres? If so, how does that impact your manuscript and outcomes? Is all the biology in question ongoing solely in APBs?
3. ATR signalling. The phospho-ATR antibody is not a well-established marker of ATR activation. Instead it is more useful to look for phosphorylation of downstream ATM targets, i.e. CHK1, RPA, etc. Further, it is not clear from the presented data that ATR is activated. Description of the western blot data/methods do not indicate if normalization to the pan-ATR antibody was used for scoring pATR abundance. From my eye it looks like ATR levels increase concordantly (if not more) than the phosphor-ATR suggesting the change simply reflects different levels of ATR in the extracts.
4. The authors evoke replication stress, but nowhere in the manuscript do they test if replication stress is involved in this mechanism. The over-expression of mCherry-RnaseH1 is nice to remove RNA:DNA hybrids, but this does not test replication stress as claimed in the text. It is possible that replication stress is part of this mechanism, but without designing experiments that directly test the involvement of DNA replication this claim is unsupported.
5. I have several concerns about the telomere length dynamics data in figure six. I do not think you can infer anything about changes in telomere length from the data in figure six. As mentioned above, it is not clear the proper statistical tests were used. Additionally, presentation as very dense dot plots present challenges to interpretation (i.e. is there any change in the median, interquartile range, distribution etc?) These data need to be presented as box and whiskers plots. Additionally, I have concerns about signal intensity. Why are the signal intensities roughly the same in ALT vs Telomerase cells, when the ALT cells have much longer telomeres? U-2OS having the longest telomeres of any ALT cell line. I would expect the signal intensity in the U-2OS cells to be multiple times greater. Additionally, the authors use very brief siRNA knockdowns (duration is directly stated in text, from methods I infer it was a three-day experiment). For telomeres to erode, the cells must go through DNA replication. It is not controlled in this experiment to know how many times the cells are dividing, and the minor differences in telomere lengths may reflect differing growth rates. Additionally, a three-day knockdown does not provide enough time for the telomeres to erode sufficiently to assay the impact of the genes in question on telomere maintenance. To do this experiment properly, the authors need to inhibit NONO and SFPQ continuously, and monitor telomere length over 50 or more population doublings. This will identify if the genes impact telomere length maintenance. Additionally, to determine if there is an impact on ALT, it would be useful to look at other ALT markers over the same duration.
6. Conclusions – The CO-FISH result with NONO and SFPQ double knockdown is lovely, and there are data of interest in this manuscript. However, the conclusions are not robustly supported by the data. For the reasons described above, there are questions regarding the outcomes and more work

is required before the conclusions are supported.

Minor experimental questions and comments.

1. Why did the authors choose to pull down proteins using a TERRA bait from mouse stem cells when the paper focuses on ALT and telomerase positive cancer cells? What is the justification?
2. Why did you perform the metaphase-TIF analysis on H1299 cells instead of U-2OS when most of the paper is focusing on ALT?
3. U-2OS is unique as the only p53-competent ALT cell line. If DNA damage at the telomeres is evoked in your system, U-2OS will react differently than other ALT lines and potentially arrest growth. In future experiments, using different ALT cells might enable you to see a greater response as p53 compromised cell lines are refractory to telomere DDR activation.

Reviewer #3 (Remarks to the Author):

Review NCOMMS-18-02957-T "SFPO and NONO suppress RNA:DNA hybrid related telomere instability"

In this manuscript by the Schoeftner laboratory the authors address the roles of NONO and SFPO in regulation of RNA/DNA hybrids at telomeres. The group shows that NONO and SFPO can interact with TERRA transcripts and suggest that the factors colocalize with telomeres. Suppression of NONO and SFPO slightly increases damage signals at telomeres and potentially elevates RNA/DNA hybrid formation at telomeres. This could result in replication stress, as an increase in localization between activated TPA32 and telomeres is observed and ATR is potentially phosphorylated. The authors suggest that the leading strand is especially affected, as Co-FISH suggests slightly more fragility there. Similarly, the group suggests that SFPO and NONO play a role in telomeric crossovers, since co-suppression of the factors strongly increases TSCE and APBs, at least in ALT cells.

Finally, the group observed small changes in telomere length upon SFPO and NONO suppression, implicating a link to recombination mediated telomere maintenance. While ALT cells increased telomere length, telomerase positive cells decreased it, leading the authors to the suggestion that loss of SFPO and NONO leads to a strong increase in recombination at telomeres.

In summary, while a few parts of the manuscript are convincing (the isolation of SFPO and NONO as telomere associated factors; the increase in TSCE upon co-suppression of SFPO and NONO), many parts-including the statistics and statistical significance- are underdeveloped and not convincing at all, and therefore this manuscript is not suitable for publication.

In detail:

A true flaw of the manuscript is the extensive use of ALT cells. Figures 1 D-G are not meaningful and the authors cannot distinguish whether the factors associate with telomeres, APBs or ECTR. The colocalization approach with γ H2AX is poor and should not be done in ALT cells (Figures 2 D, E) for the same reasons. The experiments in H1299 cells are important, but it is unclear what is going on in these cells. Telomere fragility /number of telomeric foci seems uncharacteristically high.

The colocalization between RNA/DNA hybrids and TRF1 is hard to interpret, barely elevated (Fig. 2G) and only borderline significant. Why is the TRF1 staining in the H1299 cells so uncharacteristic for telomerase positive cells? The staining looks like these were ALT cells (Figure 2H). In Figure S5 the TRF2 stain looks completely different, but should be comparable.

The RPA colocalization is accordingly difficult to interpret and the ATR phosphorylation is not convincing at all, neither by immunoblotting nor by IF. Specifically, if RNase H were to eliminate replication stress, why is the RPA32 stain in Figure 3H still elevated over the control?

The Co-FISH data are hard to evaluate without primary data. The authors suggest that a 5% to 7% increase is significant, but a 5% to 3% decrease is not? I question the statistical approach. The telomere length analysis needs primary data and southern analysis to back up the small

changes observed through Q-FISH.

Generally it would be important to use more and different cell lines, as scientific rigor and reproducibility are currently questionable. Primary data should be included in the analysis.

Also, throughout the manuscript data have been heavily overinterpreted. For example, the authors state in line 250 as conclusion from the NONO-SFPQ suppression data that " This suggests that telomere sister chromatid recombination represents an effective mechanism to resolve telomere fragility." There is simply no data presented to make this claim.

Similarly on line 257 they claim that "Depletion of NONO does not impact on APB frequency, underlining that loss of SFPQ is the major trigger for the enhancement of the ALT pathway in U-2 OS cells." Again, this is not backed up by data and such a general conclusion is misleading and not justified.

DETAILED REPLY TO REVIEWERS' COMMENTS

Petti et al. 2018

Reviewer #2 (Remarks to the Author):

In this manuscript Petti et al identify RNA interacting factors NONO and SFPQ as proteins that bind TERRA and interact with the telomere chromatin, and investigate putative roles for NONO and SFPQ in telomere biology through their TERRA associated roles. They conclude SFPQ and NONO help maintain telomere integrity through the modulation of R-loops to suppress telomere fragility and mediated homologous recombination. They also attempt to tie this activity to ALT-mediated telomere length maintenance. These are novel experiments that could influence thinking in the field, specifically regarding how RNA regulatory pathways impact telomere integrity and telomere length maintenance. While the novelty is appreciated, and their efforts are well-meaning, the manuscript unfortunately suffers from experimental and conceptual shortcomings that preclude publication. Areas of improvement are categorized into minor and major concerns, and where possible collated thematically.

Reply to Reviewer 2:

We appreciate the statement of Reviewer 2 that the data in our manuscript reports novel findings with relevance for telomere research and highlight the role of RNA regulatory pathways on telomere integrity and telomere length maintenance.

We also thank Reviewer 2 for giving us the possibility to improve the quality of data presented in the original manuscript, without requesting additional major experiments.

The revised manuscript now contains increased sample numbers and improved statistical analysis for all figures and addresses all other issues raised by reviewer 2. We hope that the manuscript is now suitable for the readership of Nature Communications.

Major thematic comments

Comment by Reviewer 2

1. Sample size. In general, for any experiment where nuclei are being analyzed, this requires a minimum of 3 experiments of 30 nuclei per experiment, for a minimal total of 90 nuclei. For metaphase analysis, a minimum of 3 experiments of 25 chromosome spreads per experiment for 75 metaphase spreads total. This impacts most of the data sets in the paper, which are typically underpowered.

Reply to Reviewer 2:

As requested by reviewer 2, we have increased sample number.

During the revision period we have improved the respective figures and have more than triplicated the number of analyzed interphase nuclei or metaphase spreads:

Revised Fig 2B, C, E, G, I – Interphase cells (min. 90 nuclei for each experimental condition)

Revised Fig 3B, D, F, H– Interphase cells (min. 90 nuclei for each experimental condition)

Revised Fig 4B-E – Metaphase spreads (min. 75 metaphase spreads for each experimental condition)

Revised Fig 5B, C, E, F – Metaphase spreads (min. 75 metaphase spreads for each experimental condition)

Revised Fig. 5H; – Interphase cells (min. 90 nuclei for each experimental condition)

New Fig. 5J; – Interphase cells (min. 90 nuclei for each experimental condition)

Revised Fig 6B, D, F, H – Interphase cells (min. 90 nuclei for each experimental condition)

Revised Supplementary figure 2F, H - Metaphase spreads (min. 45 metaphase spreads for each experimental condition) (please see justification below)

Revised Supplementary figure 3B, D - Interphase cells (min. 90 nuclei for each experimental condition)

Revised Supplementary figure 4A-D - Metaphase spreads (min. 75 metaphase spreads for each experimental condition)

New Supplementary figure 5B, D - Interphase cells (min. 90 nuclei for each experimental condition)

New Supplementary figure 2F, H: For Anti-gammaH2AX Immuno – Telomere DNA FISH experiments 45 metaphase spreads were analyzed for each experimental condition. Due to the complex preparation and delicate immunostaining/DNA FISH procedures we were not able to obtain a higher number of metaphases. Given that this experiment allows to specifically identify telomeres (in contrast to interphase cells) at chromosome ends we are sure that the results obtain are biologically relevant. Statistical relevance is supported by Mann-Whitney test.

Telomere damage data is supported by phosphorylated RPA32Ser33 and phosphorylated ATR at telomeres (**revised Fig. 3A-F, revised Supplementary figure 3A-D**). Thus, we feel that our analysis is gives important biological results, also not reaching the suggested number of 75 metaphases.

Comment by Reviewer 2

2. Statistics. My understanding is that student t-tests determine if the means of two groups are statistically different, but this relies on normally distributed data. Many of the data shown here do not appear to be normal distributions (specifically in figure 6), therefore, a t-test would be invalid. Wilcoxon-Mann-Whitney rank-sum testing is likely a more appropriate statistical test, which will determine if there are differences in the data distribution. As you are measuring larger datasets (i.e. distribution in the length of many telomeres), it is more appropriate to investigate differences in distribution as opposed to means. The data sets in question include 2A, 5B, 5C, 5E, 5F, 6A, 6B, 6E, and potentially the data in figure 4. Many of these will remain statistically significant. However, I have concerns for the quantitative data in figure 6 which does not appear to be significant.

Reply to Reviewer 2:

We thank reviewer 2 for criticism on the statistical analysis.

We now have chosen the following statistical tests for the respective figure panels (test are also indicated in the revised figure legends).

Revised Fig 1F: N=3; t-test

Revised Fig 1G: N=3; t-test

Revised Fig 2C: n>90; Wilcoxon-Mann test

Revised Fig 2B, E, G, I; N=3 (4); t-test

Revised Fig 3B, D, F, H: N=3; t-test

Revised Fig 4B, C, D, E: n>75; Wilcoxon-Mann test

Revised Fig 5B, C, E, F: n>75; Wilcoxon-Mann test

Revised Fig 5H; new Fig 5J; N=3; t-test

Revised Fig 6B, D, H; n>90; Wilcoxon-Mann test

Revised Fig 6F: N=3; t-test

Revised Supplementary Fig. 2F: n>45; Wilcoxon-Mann test

New Supplementary Fig. 2H: n>45; Wilcoxon-Mann test
Revised Supplementary Fig. 3B, D: N=3; t-test
Revised Supplementary Fig. 4A-D: n>75; Wilcoxon-Mann test
New Supplementary Fig. 5B: N=3; t-test

Comment by Reviewer 2

3. Data presentation. Bar graphs are better represented by showing individual symbols for the mean of each replicate, with the experimental mean as a line, and error bars showing standard deviation or error. Replicate variability is hidden in the bar graph format. This is appropriate for small data sets (n= 3 experiments). For larger data sets with many data points (i.e. > 50 data points) I suggest a box and whiskers graph. For example, the quantitative data in figure 2C, and 6A, B, E are hard to decipher at dot plots given the density of data points, and would be better represented with box and whiskers.

Reply to Reviewer 2:

We followed the suggestion from reviewer 2 to improve data representation in the respective figures

- In experiments where a media of 3 biological replicates (n=3) is shown, we use now bar diagrams that show standard deviation with whiskers pointing up and down. In addition, total number of analyzed cells, p-values, average and standard deviation are shown as numerically values. This change in data presentation regards:
 - Revised Fig. 1F
 - New Figure Fig. 1G
 - Revised Fig. 2B, E, G, I
 - Revised Fig. 3B, D, F, H
 - Revised Fig. 5H, New Fig. 5J
 - Revised Fig. 6F
 - Revised Supplementary figure 2B, C
 - Revised Supplementary figure 3B, D
 - Revised Supplementary figure 5B

- As requested by reviewer 2, Figures with more than 50 integrated data points are shown now in box diagrams. This change in data presentation regards:
 - Revised Fig. 2C
 - Revised Fig. 4B, C, D, E
 - Revised Fig. 5B, C, E, F
 - Revised Fig. 6B, D, H
 - Revised Supplementary figure 4B-D

- Immuno DNA-FISH data in Supplementary fig 2F, H (n=45) is shown with box blots.

Comment by Reviewer 2

4. Image quality and representation.

[A] Most of the data in this manuscript rely heavily on imaging data. There are differing problems with the data sets. Some images appear to be heavily saturated, making it difficult to interpret the outcomes (e.g. **2F, 2H, 5G, 6A, Supplementary 2H, 2I, 3A**).

Reply to Reviewer 2:

We have improved the quality of images throughout the study. These changes regard the following figures:

Revised Figure 2A, D, F, H
Revised Figure 3A, C, E, G
Revised Figure 4A
Revised Figure 5A, G
Supplementary figure 2E, G, K-M
Supplementary figure 3A, C

[B] In other data sets, co-localizations are scored, but one of the channels is diffuse puncta. For example, in **1D**, **1E**, it is hard to determine if the telomeres co-localize specifically with NONO/SFPQ or if this is a result of extensive NONO/SFPQ staining throughout the nucleus. Specific co-localization can be determined by shifting one channel several pixels, and re-counting. If the co-localizations remain similar in number after the pixel shift, it indicates these are not specific co-localizations, but simply random overlap of abundant puncta. Alternatively, for this experiment, cyto centrifugation as shown in Supplementary Figure 2D but probed with anti-NONO or SFPQ might work well to show telomere specific localization. Similar problems arise with **Figure 3A and 3C**.

Reply to Reviewer 2:

We have performed the experiment for Fig. 1D, E as suggested by reviewer 2.

We found that shifting one channel versus the other strongly reduced TRF1-SFPQ colocalization and also significantly reduced TRF2-NONO co-localization in U2-OS interphase cells. This data support that a significant fraction of nuclear NONO and SFPQ localizes to telomeres in interphase cells. This data has not been included into the revised version of the manuscript but is accessible to the reviewers and the editor of the manuscript (Figure 1 provided to reviewers).

In contrast to the strong reduction of colocalization observed TRF1-SFPQ channel shifting (ca. -3fold, $p < 0,009$), shifting of channels in TRF2-NONO co-localization is reduction is less dramatic (-18%, $p < 0,034$). We think that this is mainly caused by the different numbers of NONO and SFPQ foci observed in our immunolocalization studies. Anti-NONO antibodies reveal a much higher number of NONO foci per nucleus when compared to the number of SFPQ foci (Fig. 1D, E; Supplementary figure 1E, F, representative images provided as Figure 1 to reviewers). Consequently, the chance of having random TRF2-NONO colocalization after channel shifting is much higher compared to TRF1-SFPQ channel shifting experiments. We feel that this phenomenon contributes to the rather low (but significant) decrease of TRF2-NONO colocalization index in channel shifting experiments. However, we want to point out that the specific localization of NONO (and also SFPQ) to telomere repeat sequences is supported by telomere ChIP experiments, now performed in triplicate (revised Fig. 1G, H).

In order to provide another support for the interaction of NONO with telomere components we performed immunoprecipitation experiments. These experiments support an interaction of NONO with TRF1 and TRF2. We have added these IPs as **new Supplementary figure 1C, D** to the revised version of the manuscript

We also prepared metaphase spreads by cytospinning colcemide-blocked U2OS cells. Immunostaining for SFPQ / NONO followed by telomere DNA FISH was performed to test a localization of SFPQ or NONO to telomeres. Although big efforts were invested for this experiment, we were not able to clearly detect NONO or SFPQ at chromosome ends. Given high nuclear NONO and SFPQ protein levels we observed a punctuated pattern across the area covered by metaphase chromosomes. We also cannot exclude a possible masking of the respective SFPQ/NONO epitopes at telomeres of highly condensed metaphase chromosomes or an eventual displacement of NONO/SFPQ from telomeres in metaphase.

Considering these issues, we state in the revised version of the manuscript:

- **Page 6, line 6:** “Confocal microscopy revealed that a significant fraction of nuclear restricted SFPQ and NONO foci co-localize with telomere repeat binding factor 2 (TRF2) or telomere repeat binding factor 1 (TRF1) in U-2 OS interphase cells (Fig. 1 D-F).
- **Page 6, line 8:** We inserted the phrase: “Immunoprecipitation experiments support the interaction of NONO with TRF1 and TRF2 (Supplementary figure 2C, D).”

[C] There appears to be bleed through in the RNA:DNA hybrid staining to the telomere channel – as shown by retained TRF1 signal in Supplementary 2I following TRF1 knockdown. This brings the data on RNA:DNA hybrid imaging into question.

Reply to Reviewer 2:

We have inserted improved representative images (confocal microscopy) in the revised version of the manuscript. **(Revised Supplementary figures 2L,M)**

[D] I do not see a lagging fragile telomere in the example in 4A. The red telomere, which I infer from the diagram is supposed to be fragile, look to me to be an over-saturated telomere signal. This raises questions about data scoring of these experiments.

Reply to Reviewer 2:

We have inserted improved representative images in the revised version of the manuscript.

[E] For all data sets, it is unclear if presented images are single focal planes, maximum intensity projects, etc? This is not described but are important information.

Reply to Reviewer 2:

We thank reviewer 2 for high-lightening this issue. This information is given in the Supplementary Material and Methods section (“Microscopy”) of the revised version of the manuscript.

In particular:

- Microscope images of metaphase chromosomes subjected to DNA – FISH (and gammaH2AX immuno-FISH) are shown as single focal plane **(Revised Figure 4A; Figure 5D; Supplementary figures 2E,G)**
- Microscope images of telomere quantitative DNA – FISH on interphase cells or quantitative RNA-FISH are shown as single focal planes **(Revised Figure 6A, C, G; Figure 2A)**
- Microscope images of simple immunofluorescence co-stainings are shown and analyzed as single plane images: **revised Figure 2D; Fig. 3A, C, E; Figure 5G, I; Supplementary Figure 2K; Supplementary figure 3A, B; Supplementary figure 5A.**
- Confocal microscope images of more complex immunostainings are shown and analyzed as single focal planes. This regards the revised **Figures 1D, E; Figure 2F, H; Revised Supplementary Figures 1E, F; Revised Supplementary Figures 2L, M.**

[F] The use of arrows to identify foci is nice, but in some images the arrows are so numerous it obscures the image. It is better to show one or two examples and let the reader infer the remainder from the image.

Reply to Reviewer 2:

We have improved the figures as requested by reviewer 2.

Changes affect the following figures:

Revised Figure 1D, E

Revised Figure 2A, D, F, H

Revised Figure 3A, C, E, G

Revised Figure 5G

Revised Figure 6E
Revised Supplementary figure 2L, M
Revised Supplementary figure 3A, C
Revised Supplementary figure 5A

Major experimental comments

Comment by Reviewer 2

1. How many times was the ChIP experiment done in figure 1, and where is the quantitation? Additionally, it is appropriate to use anti-H3 as a positive control for both telomere and Alu. Currently there is no indication of the researchers ability to capability to ChIP Alu repeats, limiting its utility as a non-telomere control.

Reply to Reviewer 2:

In **revised Figure 1G, H** we show results from 3 independent ChIP experiments, including a anti-“global” histone H3 antibody. Dot blots were hybridized with a telomere repeat and a human AluY probe (see revised supplementary material and methods section (“ChIP assay and telomere dot-blots”). We clearly demonstrate that NONO and SFPQ are enriched at telomere repeats in human cancer cells. This data is shown in **revised Fig. 1G, H**.

Page 6, line 12: Performing telomere ChIP we confirm association of SFPQ and NONO with telomere repeat chromatin and exclude binding to AluY repeats (Fig. 1G, H).

Comment by Reviewer 2

For images in 2A, 2D, 2H and all of figure 3 – are you looking at telomeres or APBs? The number of TERRA or TRF2 foci are too few to be telomeres, additionally the foci are often large and bright, consistent with APBs. These experiments were done in ALT cells. Does this indicate NONO and SFPQ act at APBs instead of chromosomal telomeres? If so, how does that impact your manuscript and outcomes? Is all the biology in question ongoing solely in APBs?

Reply to Reviewer 2:

NONO/SFPQ and PML/APBs:

We did not find evidence for a localization of NONO and SFPQ to PML that represent a main component of APBs (**new Supplementary figure 1E, F**). We have inserted text related to the **new Supplementary figure 1E, F** and state:

Page 6, text line 9: “Confocal microscopy did not reveal a significant colocalization between NONO or SFPQ with PML, suggesting that the studied RNA binding proteins do not represent central components of APBs. (Supplementary Figure 1E, F)”

TERRA RNA-FISH: U2OS cells are characterized by a highly heterogeneous telomere length distribution, Accordingly, TERRA signal intensity is heterogeneous in U2OS cells. For microscopy experiments have chosen an exposure time that avoids saturation of large TERRA foci. As a consequence, the number of weaker TERRA signals is reduced. Same holds true for TRF2: in order to avoid large, saturated TRF2 signals we keeps exposure-times relatively low. Overall, we find that TERRA foci numbers/nucleus in U2-OS cell are similar to those shown in a recent study (also U2-OS cells; Arora et al 2014, PMID:25330849).

Given that the same exposure time was used for the analysis of the individual biological replica-experiments we are sure that the increase in TERRA foci number and average TERRA size is real. We improved this set of data by analyzing a minimum of 90 nuclei for each experimental condition (improved p-values). This data is shown in (**revised Fig. 2A-C**). Further we have included a **new Supplementary figure 2D** that shows the numeric distribution of TERRA foci in analyzed cells. In

fact, knock-down of NONO and SFPQ results in an increased frequency of cells with a TERRA foci-number >12.

TERRA is a component of APBs (Arora et al 2014, PMID:25330849). In the revised version of the manuscript we have also investigated co-localization of TERRA and PML. In line with increased telomere recombination frequency, we found a significant increase in TERRA-PML co-localization in siSFPQ cells (**new Supplementary figure 5A-B**). Importantly, in all conditions only a sub-fraction of TERRA foci colocalizes with PML. These data are in line with a previous study that show that TERRA accumulation is strongest in APBs linking TERRA with APBs. Of notice also in this study, only subtraction of TERRA (8%) foci shows detectable co-localization with APBs (Arora et al 2014, PMID:25330849).

Our observation that unleashing T-SCE in siSFPQ cells is linked with increased TERRA-PML co-localization suggests increased “APB-usage”

3. ATR signaling. The phospho-ATR antibody is not a well-established marker of ATR activation. Instead it is more useful to look for phosphorylation of downstream ATM targets, i.e. CHK1, RPA, etc. Further, it is not clear from the presented data that ATR is activated. Description of the western blot data/methods do not indicate if normalization to the pan-ATR antibody was used for scoring pATR abundance. From my eye it looks like ATR levels increase concordantly (if not more) than the phosphor-ATR suggesting the change simply reflects different levels of ATR in the extracts.

Reply to Reviewer 2:

We were able to show increased phospho-RPA32Ser33, phospho-gammaH2AX and phospho-ATR at telomere repeats by IF on experimental cells. Western blotting revealed phosphorylation of H2AX after a knock-down of SFPQ/NONO for 3 days. We agree with the reviewer that p-ATR was not convincingly increased under these conditions. We have performed time-course knock-down experiments to better follow the kinetics of the phosphorylation of ATR, Chk1 and RPA32Ser33. After 24 hours of NONO/SFPQ knock-down we were able to detect an increase of phosphorylation of ATR and modest increase of RPA32Ser33 in western blotting experiments (**Figure 2A, B provided to reviewers**). We also included the phosphor-Chk1 antibody in immunofluorescence stainings and western blotting experiments. However, we did not obtain usable results, presumably for technical reasons, related to the antibody-batch.

To further investigate this issue, we used immunofluorescence images from Fig.3 and quantified pan-nuclear phospho-ATR and pan-nuclear phosph-RPA32ser33 levels using ImageJ. Importantly, we found a significant increase of pan-nuclear RPA32Ser33 phosphorylation upon NONO or SFPQ depletion (**Figure 2C, provided to reviewers**). Loss of SFPQ further revealed an increase in pan-nuclear phospho-ATR levels (**Figure 2C, provided to reviewers**). In fact, results from telomeres (**revised Fig. 3**) are reproduced on the global level in experimental cells. We are confident that these results support the activation of surrogate makers of replication stress.

Our results suggest that IF is superior to western blotting in detecting p-ATR and p-RPA32Ser33. Data from gammaH2AX experiments indicate that loss of NONO or SFPQ does not induce a full-blown DNA damage response (**revised Supplementary figure. 2I, J**). We therefore propose that manipulation during extract preparation and western blotting results a significant reduction of phosphorylation of the respective targets, rendering difficult of detect an increase in p-ATR and/or p-RPA32Ser33. PFA fixation during IF appears to be a more adapted strategy to block potential phosphatases and to maintain alterations in p-ATR and p-RPA32Ser33 levels.

Due to i) differences observed between western blotting and immunofluorescence experiments and ii) the fact that new western data on RPA32Ser33 and ATR were obtained at a different timepoint (24hours) than gammaH2AX (72 hours knock-down) we decided to only include gammaH2AX western data in the revised version of the manuscript and to provide RPA32Ser33 and ATR data to

reviewers (**Figure 2, provided to reviewers**). We feel that this supports the clearness of the manuscript.

In the revised version of the manuscript we have now inserted gammaH2AX and p53 western data as **new Supplementary figure 2I, J**. At this position of the manuscript the data supports results showing gammaH2AX at telomere repeat sequences (**revised Supplementary figure 2E-H**)

Comment by Reviewer 2

4. The authors evoke replication stress, but nowhere in the manuscript do they test if replication stress is involved in this mechanism. The over-expression of mCherry-RnaseH1 is nice to remove RNA:DNA hybrids, but this does not test replication stress as claimed in the text. It is possible that replication stress is part of this mechanism, but without designing experiments that directly test the involvement of DNA replication this claim is unsupported.

Reply to Reviewer 2:

This experiment was very challenging to set up. Our data convincingly show that RNaseH1 is able to rescue RPA32-phosphorylation at telomeres triggered by loss of NONO/SFPQ. Phosphorylated RPA32 is used as robust marker for impaired replication at telomeres (Arora et al. 2014, PMID:25330849; Cox et al. 2016, PMID: 26832416).

We agree that it is very interesting to study the phenotypes observed in the context of DNA replication at telomeres. Addressing this issue would require a complex set of experiments such as replication fork dynamics studies using DNA combing. These experiments would rise a big number of new questions, that would actually go beyond the scope of this study. Thus, I feel that these issues should be better addressed in a separate study.

The aim of the current study is to report the discovery of RNA binding proteins that act as novel regulators of telomeres via the control of RNA:DNA hybrid management.

In the revised version of our manuscript we have eliminated the statement on replicative stress and write now:

- **Page 8, line 10:** Instead of previously stating:

“Induction of replicative stress at telomeres is linked with the phosphorylation of ATR and the phosphorylation of Serine 33 of the 32kDA subunit of the Replication protein A (RPA32 pSer33).”

We state now:

“Induction of replication defects at telomeres is linked with the phosphorylation of ATR and the phosphorylation of Serine 33 of the 32kDA subunit of the Replication protein A (RPA32pSer33), both surrogate markers for replication stress”

- **Page 9, line 5:** Instead of previously stating:

“We next wished to test whether increased RNA:DNA hybrid formation is directly linked to replicative stress in NONO/SFPQ depleted cells. “

We state now:

“We next wished to test whether increased RNA:DNA hybrid formation in NONO/SFPQ depleted cells is linked to phosphorylation of RPA32.”

- **Page 9, line 6:** Instead of previously stating:

“To address this issue, we aimed to rescue replicative stress in NONO and SFPQ depleted...”

We state now:

“To address this issue we aimed to rescue RPA32 phosphorylation levels in NONO and SFPQ depleted...”

- **Page 9, line 11:** Instead of previously stating:

“...preventing RNA:DNA hybrid accumulation and R-loop related replicative stress at telomeres.”

We state now:

“...preventing RNA:DNA hybrid accumulation and R-loop related replication defects at telomeres.”

- **Page 15, line 29:** Instead of previously stating:

“In line with the induction of replication stress we observed increased loading of the classic DNA damage marker γ H2AX at telomeres of NONO or SFPQ depleted cells.”

We state now:

“In line with the increased abundance of RNA:DNA hybrids we observed increased loading of the classic DNA damage marker γ H2AX at telomere repeats of NONO or SFPQ depleted cells.”

- **Page 15, line 17:**

In order to address concerns of reviewer 2 on the definition of the term “replication stress” we have exchanged the words “replicative stress” (original version) for “impaired replication” (revised version)

Comment by Reviewer 2

5. I have several concerns about the telomere length dynamics data in figure six.

(A) I do not think you can infer anything about changes in telomere length from the data in figure six. As mentioned above, it is not clear the proper statistical tests were used. Additionally, presentation as very dense dot plots present challenges to interpretation (i.e. is there any change in the median, interquartile range, distribution etc?) These data need to be presented as box and whiskers plots.

(B) Additionally, I have concerns about signal intensity. Why are the signal intensities roughly the same in ALT vs Telomerase cells, when the ALT cells have much longer telomeres? U-2OS having the longest telomeres of any ALT cell line. I would expect the signal intensity in the U-2OS cells to be multiple times greater.

(C) Additionally, the authors use very brief siRNA knockdowns (duration is directly stated in text, from methods I infer it was a three-day experiment). For telomeres to erode, the cells must go through DNA replication. It is not controlled in this experiment to know how many times the cells are dividing, and the minor differences in telomere lengths may reflect differing growth rates. Additionally, a three-day knockdown does not provide enough time for the telomeres to erode sufficiently to assay the impact of the genes in question on telomere maintenance. To do this experiment properly, the authors need to inhibit NONO and SFPQ continuously, and monitor telomere length over 50 or more population doublings. This will identify if the genes impact telomere length maintenance. Additionally, to determine if there is an impact on ALT, it would be useful to look at other ALT markers over the same duration.

Reply to Reviewer 2:

A) As requested by reviewer 2 we show now Q-FISH data in box blot diagrams that indicate mean telomere length and whiskers showing standard deviation. We performed a Mann-Whitney test to confirm statistic relevance.

In ALT cells single telomeres can undergo dramatic length fluctuations during cycles of cell division, with rapid shortening and rapid lengthening events (Murnane et al. 1994 PMID: 7957062; Perrem et al. 2001, PMID: 11359895). Given that SFPQ deletion in telomerase negative U2-OS cells increases T-SCE and APB frequency we conclude that loss of SFPQ can

enhance the ALT pathway to elongate telomeres in the shown 3-day period. On the contrary, telomerase positive, ALT incompetent, SFPQ knock-down H1299 cells show increased T-SCE rate, but no increase in APB numbers and exhibit reduced telomere length. Only stimulating the ALT pathway by 5'aza treatment, as exemplified by increased APB numbers, allows to translate increased T-SCE rates into telomere elongation in SFPQ loss of function H1299 cells. Thus, we are convinced that short term loss of function experiments gives a clear information on the principal and mechanistic function of SFPQ in the ALT pathway. Such a role is also supported by SFPQ loss of function mutations in osteosarcoma, that normally use ALT to maintain telomere function (Kovac et al. 2015, PMID: 26632267).

- B) H1299 and U2OS cells show profoundly different medium telomere length but also different telomere length distribution. H1299 cells have a medium telomere length of about 3 kb; instead U2OS cells have medium telomere length of ca. 35kb and as a result of ALT, telomeres can be very short but also very long (up to 100kb) (Huang et al. 2017, PMID: 28366536; Min et al 2017; PMID: 28760773)

When doing quantitative DNA-FISH, saturation of telomere signals needs to be avoided. Accordingly, exposure times of telomere DNA-FISH are different between U2-OS (longer telomeres; shorter exposure time) and H1299 (shorter telomeres, longer exposure time) cells. This finally causes that average a.f.u of H1299 and U2-OS telomeres appear in the same range, as shown in **revised Figure 6**.

- C) **Revised Figure 6A** shows a significant increase in telomere length in siSFPQ cells in a short time period of 3-day knock-down conditions. These rapid changes in telomere length are only possibly due to the induction of telomere repeat recombination events (see answer to point A). In this study we only look at “immediate early” effects of loss of SFPQ function, by analyzing short experimental time-windows. Presumably due to the pleiotropic functions of SFPQ in cell metabolism, cells lacking SFPQ show impaired cell proliferation. Thus, long-term experiments would be difficult to perform. Currently we are generating cancer related NONO/SFPQ point. These mutations are expected to do not have impact on cancer cell proliferation.

In the revised version of the manuscript we highlight that we are investigating the immediate consequences of loss of SFPQ/NONO on telomere stability. We now state:

- **Page 9, line 19:**

“In order to understand the direct importance of NONO and SFPQ for telomere integrity we performed short term loss of experiments using telomere chromosome orientation DNA-FISH (CO-FISH).”

- **Page 16, line 10**

“We thus conclude that NONO and SFPQ have a direct role in suppressing RNA:DNA hybrid related telomere fragility and recombination (Fig. 7)”

Comment by Reviewer 2

6. Conclusions – The CO-FISH result with NONO and SFPQ double knockdown is lovely, and there are data of interest in this manuscript. However, the conclusions are not robustly supported by the data. For the reasons described above, there are questions regarding the outcomes and more work is required before the conclusions are supported.

Reply to Reviewer 2:

In a major effort we have significantly increased the number of metaphase chromosome spreads analyzed in the different experiments (n=75) and have also increased the number of analyzed interphase nuclei (>90). Summarizing we have:

- triplicated the number of analyzed cells/metaphases in individual figures.
- performed a panel of anti-NONO and anti-SFPQ ChIP experiments (**Revised Fig. 1G**)
- validated the localization of NONO and SFPQ at telomeres (**Figure 1, provided to reviewer**)
- we further show new data on the recruitment of RAD51 to telomeres in SFPQ knock-down cells (**New Figure 5I, J**)
- address the localization of TERRA to PML bodies (**new Supplementary figure 5A, B**)
- show telomere damage in metaphase spreads of SFPQ knock-down U2OS cells (**new Supplementary figure 3E,F**).
- statistical analysis was re-done and give large improved p-values.
- as requested, conclusions were adjusted and rephrased.

Thus, we are now confident data in our manuscript supports the following conclusions:

- 1.SFPQ and NONO are TERRA binding proteins that interact with telomere repeats
- 2.SFPQ and NONO suppress RNA:DNA hybrid formation at telomeres
- 3.Loss of SFPQ and NONO mediate replication defects at telomeres
- 4.Loss of SFPQ and NONO mediate telomere damage
- 4.NONO suppresses telomere fragility
- 5.SFPQ suppresses homologous recombination at telomeres
6. Loss of SFPQ and NONO induces high telomere recombination frequency and increased number of APBs
7. Loss of SFPQ promote telomere elongation in recombination competent ALT cells

Comment by Reviewer 2

Minor experimental questions and comments.

1. Why did the authors choose to pull down proteins using a TERRA bait from mouse stem cells when the paper focuses on ALT and telomerase positive cancer cells? What is the justification?

Reply to Reviewer 2:

In the initial phase of the project we aimed to identified TERRA binding proteins in a classic cell model that has long telomeres, show good TERRA expression, defined chromatin structure and shows active telomerase and ALT dependent telomere maintenance mechanisms. Thus, mouse embryonic stem cells (telomerase positive; average 4% T-SCE frequency) were the cell line of choice. (**Fig. 1A, B, C**). TERRA interaction of candidates was validated using a human cancer cell line (**Fig. 1C**).

Page 5, text-line 9: we inserted a justification why mouse embryonic stem cell extract were used for RNA pull down experiments; Reference 39 refers to a work that demonstrates T-SCE and telomerase activity in mESCs.

“using mouse embryonic stem cells that maintain telomeres via telomerase dependent and independent telomere maintenance pathways³⁹”

Comment by Reviewer 2

2. Why did you perform the metaphase-TIF analysis on H1299 cells instead of U-2OS when most of the paper is focusing on ALT?

Reply to Reviewer 2:

We have now performed gammaH2AX – Telomere DNA FISH experiments in U2-OS cells. In line with data from revised Fig. 2D, E and revised Fig. 3C-F we show that loss of SFPQ leads to telomere damage in U2-OS cells (**New supplementary figure 2G,H**).

Page 7, text line 10: “Analysis of metaphase chromosomes by immune-DNA-FISH showed increased localization of γ H2AX at telomere sequences at chromosome ends in H1299 and U-2 OS cells (Supplementary figure 2E, H).

Page 7, line 12: We removed western data from main Figure 3 and shifted gammaH2AX western blots from U-2 OS and 1299 cells to the Supplementary figure 2I, J. We state now:

“Activation of a DNA damage response was validated by western blotting as shown by increased γ H2AX levels and stabilization of p53 in U-2 OS cells.”

Comment by Reviewer 2

3. U-2OS is unique as the only p53-competent ALT cell line. If DNA damage at the telomeres is evoked in your system, U-2OS will react differently than other ALT lines and potentially arrest growth. In future experiments, using different ALT cells might enable you to see a greater response as p53 compromised cell lines are refractory to telomere DDR activation.

Reply to Reviewer 2:

As mentioned in the reply to comment 5C we observe defects in cell proliferation in both p53 proficient U2-OS but also p53 null H1299 SFPQ knock down cells. Thus, the effect on cell proliferation is not direct linked to the p53 status and might be linked to the pleiotropic function of SFPQ in RNA metabolism. We are currently generating SFPQ point mutations (occurring in osteosarcoma) to circumvent this problem.

Reviewer #3 (Remarks to the Author):

Review NCOMMS-18-02957-T "SFPQ and NONO suppress RNA:DNA hybrid related telomere instability"

In this manuscript by the Schoeftner laboratory the authors address the roles of NONO and SFPQ in regulation of RNA/DNA hybrids at telomeres. The group shows that NONO and SFPQ can interact with TERRA transcripts and suggest that the factors colocalize with telomeres. Suppression of NONO and SFPQ slightly increases damage signals at telomeres and potentially elevates RNA/DNA hybrid formation at telomeres. This could result in replication stress, as an increase in localization between activated TPA32 and telomeres is observed and ATR is potentially phosphorylated.

The authors suggest that the leading strand is especially affected, as Co-FISH suggests slightly more fragility there. Similarly, the group suggests that SFPQ and NONO play a role in telomeric crossovers, since co-suppression of the factors strongly increases TSCE and APBs, at least in ALT cells.

Finally, the group observed small changes in telomere length upon SFPQ and NONO suppression, implicating a link to recombination mediated telomere maintenance. While ALT cells increased telomere length, telomerase positive cells decreased it, leading the authors to the suggestion that loss of SFPQ and NONO leads to a strong increase in recombination at telomeres.

In summary, while a few parts of the manuscript are convincing (the isolation of SFPQ and NONO as telomere associated factors; the increase in TSCE upon co-suppression of SFPQ and NONO), many parts-including the statistics and statistical significance- are underdeveloped and not convincing at all, and therefore this manuscript is not suitable for publication.

Reply to Reviewer 3:

We thank reviewer 3 for stating that the manuscript contains convincing results related to the discovery of NONO/SFPQ as novel telomere associated factors and also data on telomere alterations in NONO/SFPQ loss of function.

We have invested a big effort in improving the solidness of our data by performing new experiments and by triplicate the number of analyzed sample for almost all experiments shown.

We hope that this improvement and all the other introduced modifications and new data address the concerns of reviewer3.

Comment by Reviewer 3

In detail:

A true flaw of the manuscript is the extensive use of ALT cells. Figures 1 D-G are not meaningful and the authors cannot distinguish whether the factors associate with telomeres, APBs or ECTR.

Reply to Reviewer 3:

- In the revised version of the manuscript we analyzed co-localization of PML bodies with NONO and SFPQ by confocal microscopy (**New Supplementary figure 1E, F**). In this setup SFPQ and NONO appear to be scattered throughout the nucleus (but excluded from nucleoli). We did not find evidence for a co-localization pattern between NONO and PML, or SFPQ and PML. Given that PML is a main component of APBs, this data so far does not support a key role of NONO and SFPQ in APBs. Further, we did not find reports linking NONO/SFPQ with APBs.
- Further, we consolidated ChIP and Immunofluorescence data showing that NONO and SFPQ localize to telomere repeats: Presented ChIP data (**revised Fig. 1D - F**) was performed in triplicate (n=3) and a global histone H3 antibody was included. IF experiments indicate colocalization of TRF1 with SFPQ and NONO with TRF2 (**revised Fig. 1D,E**). Importantly, shifting the TRF1 or TRF2 channel in TRF1-SFPQ and TRF2-NONO co-localization studies result a reduced colocalization index, as determined by ImageJ analysis (**Figure 1 provided to reviewers**). Finally

we show by immunoprecipitation that NONO interacts with TRF1 and TRF2 (**new Supplementary figure 1C,D**)

- Phenotypes in NONO and SFPQ loss of function cells are clearly detectable at telomere repeats located on chromosome ends, as determined by CO-FISH (Fragility, T-SCE; **revised Figures 4 and 5**), Immuno-Telomere DNA FISH (DNA damage at telomeres; **new and revised Supplementary Figure 2E, G**) and telomere length alterations (**revised Figure 6A-E**). Thus, we conclude that NONO and SFPQ have a function in controlling telomeres.

We agree with the reviewer that we cannot exclude that NONO or SFPQ may also function at ECTRs. However:

- i) to my knowledge, ECTRs have so far not been reported to be transcribed or to give rise to RNA:DNA hybrids
- ii) we found clear RNA:DNA hybrid related phenotypes at telomeres of metaphase chromosomes (fragility, T-SCE, DNA damage) and
- iii) telomerase positive H1299 cells that do not form ECTRs reproduce phenotypes of U2OS loss of function experiments.

To our opinion, this list of arguments support a role for NONO and SFPQ at telomeres.

We feel that it is very important to include the ECTR argument into the discussion section of the revised version of the manuscript. Thus, we state now:

Page 14, line 24: In ALT cells, extra-chromosomal telomeric repeats (ECTR) can constitute a notable fraction of the total telomere repeat content, suggesting that a fraction of NONO and SFPQ may also locate to this type of telomere repeat sequences⁵¹. However, alterations at telomeres of metaphase chromosomes observed in loss of function experiments support a role for NONO and SFPQ at telomeres. NONO appears to suppress telomere fragility with preference for the leading, CCCTAA repeat containing telomeric strand.

- The use of U2OS ALT cells in this study is justified by the fact that several studies have shown that RNA:DNA hybrids promote the formation of recombinogenic telomere repeat sequences that are “picked up” by the ALT pathway (Arora et al 2014, PMID:25330849; Maicher et al. 2012, PMID: 22553368; Balk et al. 2013, PMID: 24013207; Yu et al. PMID: 24550456;

An additional support for the use of ALT cells comes from a study that reports SFPQ mutations in human osteosarcoma that preferentially use the ALT pathway for telomere maintenance (Kovac et al. 2015, PMID: 26632267). Thus, we feel that U2OS ALT cells are suitable for this study. However, it was also central to us to show that the molecular events caused by loss of NONO/SFPQ also appear in the context of telomerase positive human cancer cells that do not employ the ALT pathway. In fact, telomerase positive H1299 cells largely recapitulate telomere phenotypes of U2OS cells.

Our study highlights the importance of RNA:DNA hybrids in feeding the ALT pathway in telomerase negative cancer cells that have the genetic makeup to efficiently perform recombination supported by ABPs.

To make these issues clear to the reader we have inserted the following modifications onto the manuscript:

- **Page 6, line 12:** We slightly re-phrased the conclusion of ChIP experiments. We state now: “Performing telomere ChIP we confirm association of SFPQ and NONO with telomere repeat chromatin and exclude binding to AluY repeats (Fig. 1G, H).”
- **Page 6, text line 9:** we have inserted text related to the **new Supplementary figure 1E, F:** “Confocal microscopy did not reveal a significant colocalization between NONO or SFPQ with

PML, suggesting that the studied RNA binding proteins do not represent central components of ABPs. (Supplementary Figure 1E, F)”

- **Page 6, text line 14:** We are stating now that: “SFPQ and NONO represent TERRA interacting proteins that localize to telomere repeat sequences.
- **Page 14, line 19:** We exchanged “novel telomere associated proteins” for “novel telomere repeat associated proteins”
- **Page 14, line 24** We added a comment that introduces the possibility that ECTRs may also be associated with SFPQ/NONO. “In ALT cells, extra-chromosomal telomeric repeats (ECTR) can constitute a notable fraction of the total telomere repeat content, suggesting that a fraction of NONO and SFPQ may also locate to this type of telomere repeat sequences⁵¹. However, telomere alterations of metaphase chromosomes observed in loss of function experiments support a role for NONO and SFPQ at telomeres.”
- **Page 16, line 10:** Thus, NONO and SFPQ have a direct role in suppressing RNA:DNA hybrid related telomere fragility and recombination (Fig. 7)

Comment by Reviewer 3

The colocalization approach with γ H2AX is poor and should not be done in ALT cells (Figures 2 D, E) for the same reasons.

Reply to Reviewer 3:

We improved experiments on TERRA- γ H2AX colocalization in interphase cells: **revised Figure 2D, E** contains now data from more than 90 nuclei, substantially improving the robustness and statistical significance of the obtained results.

Telomeres in ALT cells are prone to form spontaneous and chronic DNA damage (Cesare et al., 2009, PMID: 19935685; Lovejoy et al., 2012, PMID: 22829774). To further strengthen telomere DNA damage data, we performed γ H2AX IF combined with telomere DNA FISH on metaphase spreads of U2OS and H1299 cells that were previously depleted for NONO and SFPQ. In this setup we were clearly able to observe increased γ H2AX at chromosome ends in experimental cells compared to control cells (**Revised and New Supplementary figure 2E-H**).

Our data show increased telomere DNA damage under loss of function conditions. We state now: **Page 7, line 10:** “Analysis of metaphase chromosomes by immuno DNA-FISH showed increased localization of γ H2AX at telomere sequences at chromosome ends in H1299 and U-2 OS cells (Supplementary figure 2E-H). Together, this indicates that loss of NONO and SFPQ leads to altered TERRA homeostasis and promotes the formation of DNA damage at telomeres.”

Comment by Reviewer 3

The experiments in H1299 cells are important, but it is unclear what is going on in these cells. Telomere fragility /number of telomeric foci seems uncharacteristically high.

Reply to Reviewer 3:

A recent study of the Shay group addressed telomere fragility in H1299 cells. In this study, the authors show 4-5% telomere fragility – thus comparable with our telomere fragility values (Min et al 2017; PMID: 28760773).

We want to underline that the telomere fragility data in **revised Figure 4** and **revised Supplementary Fig. 4** was generated in separate experiments; thus, we were not able to exclude slight experiment-to-experiment variations.

Comment by Reviewer 3

The colocalization between RNA/DNA hybrids and TRF1 is hard to interpret, barely elevated (Fig. 2G) and only borderline significant.

Reply to Reviewer 3:

In the revised version we have **repeated the experiment** and increased (almost triplicated) the total number of nuclei analyzed (n>90) (**revised Figure 2F-I**). These new experiments, performed with confocal microscopy, confirmed the original experiments shown in initial version of the manuscript. A Man-Whitney test was used for statistical analysis of the data. We want to point out to the reviewer that we had to employ rather harsh conditions to provide access of the S9.6 anti-RNA:DNA hybrid antibody to its epitope (MeOH and Acetone). Naturally, this reduced the quality of the TRF1 staining. This presumably led to an underestimation of the co-localization frequencies. Although testing different approaches, we were so far not able to further improve staining techniques.

However, given that both rounds of experiment resulted a statistical relevant increase in RNA:DNA hybrids at telomere repeats that are coupled with classic outcomes of RNA:DNA hybrids (increased T-SCE frequency, DNA damage, recruitment of markers of impaired DNA replication, fragility) we are confident that our data are biologically relevant.

Comment by Reviewer 3

Why is the TRF1 staining in the H1299 cells so uncharacteristic for telomerase positive cells?

The staining looks like these were ALT cells (Figure 2H).

Reply to Reviewer 3:

We thank reviewer 3 for pointing out this issue. In the revised version we included new representative, confocal images of TRF1 and RNA:DNA hybrid staining (**revised Fig. 2H, Supplementary Fig. 2L, M**).

Comment by Reviewer 3

In Figure S5 the TRF2 stain looks completely different, but should be comparable. The RPA colocalization is accordingly difficult to interpret and the ATR phosphorylation is not convincing at all, neither by immunoblotting nor by IF.

Reply to Reviewer 3:

Figure S5 (=Supplementary figure 5??) shows a TRF2-PML staining. Thus, I suppose that the comment of reviewer 3 refers to Figures 3C, E and/or Supplementary Figure 3A, C.

Immunofluorescence data: We thank reviewer 3 to highlight this issue. We have replaced representative images; further we show zoom images of co-localization events in the respective images. F

Western blotting:

We were able to show increased phospho-RPA32Ser33, phospho-gammaH2AX and phospho-ATR at telomere repeats by IF on experimental cells. Western blotting revealed phosphorylation of H2AX after a knock-down of SFPQ/NONO for 3 days. We agree with the reviewer that p-ATR was not convincingly increased under these conditions. We have performed time-course knock-down experiments to better follow the kinetics of the phosphorylation of ATR, Chk1 and RPA32Ser33. After 24 hours of NONO/SFPQ knock-down we were able to detect an increase of phosphorylation of ATR and modest increase of RPA32Ser33 in western blotting experiments (**Figure 2A, B provided to reviewers**). We also included the phosphor-Chk1 antibody in immunofluorescence stainings and western blotting experiments. However, we did not obtain usable results, presumably for technical reasons, related to the antibody-batch.

To further investigate this issue, we used immunofluorescence images from Fig.3 and quantified pan-nuclear phospho-ATR and pan-nuclear phosph-RPA32ser33 levels using ImageJ. Importantly, we found a significant increase of pan-nuclear RPA32Ser33 phosphorylation upon NONO or SFPQ

depletion (**Figure 2C, provided to reviewers**). Loss of SFPQ further revealed an increase in pan-nuclear phospho-ATR levels (**Figure 2C, provided to reviewers**). In fact, results from telomeres (**revised Fig. 3**) are reproduced on the global level in experimental cells. We are confident that these results support the activation of surrogate markers of replication stress.

Our results suggest that IF is superior to western blotting in detecting p-ATR and p-RPA32Ser33. Data from gammaH2AX experiments indicate that loss of NONO or SFPQ does not induce a full-blown DNA damage response (**revised Supplementary figure. 2I, J**). We therefore propose that manipulation during extract preparation and western blotting results a significant reduction of phosphorylation of the respective targets, rendering difficult to detect an increase in p-ATR and/or p-RPA32Ser33. PFA fixation during IF appears to be a more adapted strategy to block potential phosphatases and to maintain alterations in p-ATR and p-RPA32Ser33 levels.

Due to i) differences observed between western blotting and immunofluorescence experiments and ii) the fact that new western data on RPA32Ser33 and ATR were obtained at a different timepoint (24hours) than gammaH2AX (72 hours knock-down) we decided to only include gammaH2AX western data in the revised version of the manuscript and to provide RPA32Ser33 and ATR data to reviewers (**Figure 2, provided to reviewers**). We feel that this supports the clearness of the manuscript.

In the revised version of the manuscript we have now inserted gammaH2AX and p53 western data as **new Supplementary figure 2I, J**. At this position of the manuscript the data supports results showing gammaH2AX at telomere repeat

Comment by Reviewer 3

Specifically, if RNase H were to eliminate replication stress, why is the RPA32 stain in Figure 3H still elevated over the control?

Reply to Reviewer 3:

We thank reviewer 3 for pointing out this issue. In the revised version we have exchanged representative images in the **revised version of Fig. 3H**.

Comment by Reviewer 3

The Co-FISH data are hard to evaluate without primary data. The authors suggest that a 5% to 7% increase is significant, but a 5% to 3% decrease is not? I question the statistical approach.

Reply to Reviewer 3:

We have dramatically increased the number of analyzed metaphase chromosomes (Mann-Whitney test) and show now 75 metaphase spreads for every experimental condition in CO-FISH experiments. Performing these experiments, we have substantially improved statistical significance values. The improvements regard data shown in the **revised figures Fig. 4; Fig. 5 and revised Supplementary figure 4**.

We are now convinced that the revised version of the manuscript shows biological and statistically significant and reliable data.

Comment by Reviewer 3

The telomere length analysis needs primary data and southern analysis to back up the small changes observed through Q-FISH.

Reply to Reviewer 3:

We increased the number of analyzed interphase nuclei (n=90). We observe a significant telomere elongation phenotype already after 3 day-depletion of SFPQ in U2-OS cells that is even higher (+34%) in NONO/SFPQ co-depleted cells, that show high T-SCE frequency (**revised Figure 6A; revised Figure 5**). In ALT cells single telomeres can undergo dramatic length fluctuations during cycles of cell

division, with rapid shortening and rapid lengthening events (Murnane et al. 1994 PMID: 7957062; Perrem et al. 2001, PMID: 11359895). Given that SFPQ deletion (or SFPQ/NONO co-depletion) in telomerase negative U2OS cells increases T-SCE and APB frequency, we conclude that loss of SFPQ pushes recombination events via the ALT pathway to elongate telomeres, even in short 3-day period. Importantly, telomerase positive “ALT incompetent” H1299 cells do not show telomere elongation upon loss of SFPQ. Only stimulating the ALT pathway by 5-Aza-2'-deoxycytidin treatment allows to translate increased recombination into telomere elongation. Thus, we are convinced that short term loss of function experiments give a clear information on the basic and molecular role of SFPQ in the ALT pathway. The importance of SFPQ is also supported by SFPQ loss of function mutations in osteosarcoma, that normally uses ALT to maintain telomere function (Kovac et al. 2015, PMID: 26632267).

Quantitative Q-FISH is a stable method in the laboratory that gives reliable information on the length distribution of individual telomeres and its resolution is superior to Telomere Restriction Fragment analysis (TRF); especially when dealing with smaller telomere length changes and short term experiments. Further, telomere length heterogeneity in observed in U2-OS cells (medium length: 35kb; max length <100kb; Huang et al. 2017, PMID:28366536) renders TRF less applicable for the scope of the study.

We therefore decided to use quantitative telomere Q-FISH on interphase cells.

I am not clear about the usefulness of primary data from DNA-FISH experiments – however, if requested, we can provide excel sheets showing arbitrary fluorescence units for each telomere analyzed (on average 5000 values per experimental sample).

Comment by Reviewer 3

Generally, it would be important to use more and different cell lines, as scientific rigor and reproducibility are currently questionable. Primary data should be included in the analysis.

Reply to Reviewer 3:

Experiments of the manuscript have been carried in U2-OS and H1299 cell, classic human cancer cell lines that cover the telomerase independent (U2OS) and telomerase dependent (H1299) telomere maintenance pathway, respectively. These cells were used to demonstrate the general relevance of NONO and SFPQ in cancer cells. Future experiments will focus on panels of more defined human cancer cells, in particular osteosarcoma cell lines with defined oncogenic driver mutations (Kovac et al. 2015, PMID: 26632267).

I want to line out that we have substantially increased the robustness of our data from both cell lines in the revised version of our manuscript. The number of analyzed experimental samples were increased in virtually all figures. In particular, we increased the number of nuclei, analyzed by immunofluorescence staining to >90. Further we increase the number of metaphase spreads analyzed by CO-FISH to >75. This resulted substantially improved p-values (Mann-Whitney test) that further underline the robustness and biological relevance of our data.

Primary data would mean long excel lists with data from individual telomeres/chromosomes. This information is actually now nicely visualized using box-blots. If specifically requested, we are happy to provide the lists with all the raw data (primary data).

Comment by Reviewer 3

Also, throughout the manuscript data have been heavily overinterpreted. For example, the authors state in line 250 as conclusion from the NONO-SFPQ suppression data that “This suggests that telomere sister chromatid recombination represents an effective mechanism to resolve telomere fragility.” There is simply no data presented to make this claim.

Similarly, on line 257 they claim that “Depletion of NONO does not impact on APB frequency, underlining that loss of SFPQ is the major trigger for the enhancement of the ALT pathway in U-2 OS cells.” Again, this is not backed up by data and such a general conclusion is misleading and not justified.

Reply to Reviewer 3:

We have carefully gone through the manuscript to evaluate scientific conclusions. In the revised version of our manuscript we have corrected the following statements:

- **Page 6, line 6:** We slightly re-phrased the conclusion of NONO/SFPQ – telomere localization experiments. We state now: “Confocal microscopy revealed that a significant fraction of nuclear restricted SFPQ and NONO foci co-localize with telomere repeat binding factor 2 (TRF2) or telomere repeat binding factor 1 (TRF1) in U-2 OS interphase cells (Fig. 1 D-F).”
- **Page 7, line 14:** We change conclusion from: “Together, this indicates that NONO and SFPQ suppress TERRA:telomere RNA:DNA hybrid formation in telomerase positive and negative cancer cells.” to: “Together, this indicates that NONO and SFPQ have a role in suppressing TERRA:telomere RNA:DNA hybrid formation in telomerase positive and negative cancer cells.”
- We deleted the phrase from the **original version, Page 10, line 225:** “This suggests that loss of SFPQ results in the release of mechanisms that may “repair” telomere fragility. At this point of the manuscript this conclusion is not adequate.
- **Original version (line 250):** “This suggests that telomere sister chromatid recombination represents an effective mechanism to resolve telomere fragility.”
Page 11, line 17 revised version: “This suggests that telomere sister chromatid recombination may represent an effective mechanism to resolve telomere fragility triggered by NONO depletion.”
- **Original version (line 257):** “Depletion of NONO does not impact on APB frequency, underlining that loss of SFPQ is the major trigger for the enhancement of the ALT pathway in U-2 OS cells.”
Page 12, line 4 revised version: “As expected, depletion of NONO does not impact on APB frequency in U-2 OS cells.”
- **Page 15, line 17:** We introduced the following change of conclusion:
Original version of manuscript:
“NONO appears to have a selective role in preventing fragility of the leading, CCCTAA repeat containing telomeric strand
Revised version of manuscript:
“NONO appears to suppress telomere fragility with preference for the leading, CCCTAA repeat containing telomeric strand”
- **Page 15, line 24:** We introduced the following change of conclusion:
Original version of manuscript:
“This effect is paralleled by a reduction of telomere fragility, indicating that homologous recombination can rescue telomere fragility. This indicates that SFPQ functions as an important barrier to homologous recombination at telomeres.”
Revised version of manuscript:
“This effect is paralleled by a reduction of telomere fragility, leading to the interesting speculation that homologous recombination may rescue telomere fragility. Our data show that SFPQ functions as barrier to homologous recombination at telomeres”
- **Page 16, line 21** We modified the following phrase by replacing “are required” to “are important”:
“This suggests that functional APBs are important to convert increased T-SCE frequencies in SFPQ loss of function cells into an overall increase in telomere length”.

List of changes to the manuscript:

Petti et al. 2018

Introduction section:

Page 2, line 8; Page 4, line 19: “replication stress” was replaced by “DNA replication defects”

Results section:

Page 5, line 9: we inserted a justification why mouse embryonic stem cell extracts were used for RNA pull down experiments: “using mouse embryonic stem cells that maintain telomeres via telomerase dependent and independent telomere maintenance pathways³⁹. Reference 39 refers to a work that demonstrates T-SCE and telomerase activity in mESCs”

Page 5, line 16: we corrected a grammatical error: “a large set”

Page 6, line 6: We slightly re-phrased the conclusion of NONO/SFPQ – telomere localization experiments. We state now: “Confocal microscopy revealed that a significant fraction of nuclear restricted SFPQ and NONO foci co-localize with telomere repeat binding factor 2 (TRF2) or telomere repeat binding factor 1 (TRF1) in U-2 OS interphase cells (Fig. 1 D-F).”

Page 6, line 8: We inserted the phrase: “Immunoprecipitation experiments support the interaction of NONO with TRF1 and TRF2 (Supplementary figure 2C, D).”

Page 6, text line 9: we have inserted text related to the **new Supplementary figure 1C, D**: “Confocal microscopy did not reveal a significant colocalization between NONO or SFPQ with PML, suggesting that the studied RNA binding proteins do not represent central components of ABPs. (Supplementary Figure 1C, D)”

Page 6, line 12: We slightly re-phrased the conclusion of ChIP experiments. We state now: “Performing telomere ChIP we confirm association of SFPQ and NONO with telomere repeat chromatin and exclude binding to AluY repeats (Fig. 1G, H).”

Page 6, text line 13: Considering the concern of reviewer 3 on the type of telomere repeats bound by SFPQ and NONO, we are stating now that: “SFPQ and NONO represent TERRA interacting proteins that localize to telomere repeat sequences.”

Page 7, text line 4: we inserted text referring to new Supplementary figure 2C: “In line with this, the proportion of cells with high TERRA foci number (>12) was increased in experimental cells (Supplementary figure 2D).”

Page 7, text line 10: we inserted text referring to new Supplementary figure 2C: “Analysis of metaphase chromosomes by immune-DNA-FISH showed increased localization of γ H2AX at telomere sequences at chromosome ends in H1299 and U-2 OS cells (Supplementary figure 2E, EH).”

Page 7, line 10: We removed western data from main Figure 3 and shifted gammaH2AX western blots from U-2 OS and 1299 cells to the Supplementary figure 2I, J. We state now:

“Activation of a DNA damage response was validated by western blotting as shown by increased γ H2AX levels and stabilization of p53 in U-2 OS cells.”

Page 7, line 14: Considering the concern of reviewer 3 we decided to change conclusion from:” Together, this indicates that NONO and SFPQ suppress TERRA:telomere RNA:DNA hybrid formation in telomerase positive and negative cancer cells.” To:

“Together, this indicates that loss of NONO and SFPQ leads to altered TERRA homeostasis and promotes the formation of DNA damage at telomeres.”

Page 8, line 12: Addressing the concern of reviewer 2 on the definition of replication stress we now do not directly link phosphorylation of RPA32 (and p-ATR) to replication stress.

Instead of previously stating:

“Induction of replicative stress at telomeres is linked with the phosphorylation of ATR and the phosphorylation of Serine 33 of the 32kDA subunit of the Replication protein A (RPA32pSer33).”

We state now:

“Induction of replication defects at telomeres is linked with the phosphorylation of ATR and the phosphorylation of Serine 33 of the 32kDA subunit of the Replication protein A (RPA32pSer33), both surrogate markers for replication stress”

Page 9, line 5: Addressing the concern of reviewer 2 on the definition of replication stress we introduced the following change:

Instead of previously stating:

“We next wished to test whether increased RNA:DNA hybrid formation is directly linked to replicative stress in NONO/SFPQ depleted cells. “

We state now:

“We next wished to test whether increased RNA:DNA hybrid formation in NONO/SFPQ depleted cells is linked to phosphorylation of RPA32.”

Page 9, line 6: Addressing the concern of reviewer 2 on the definition of replication stress we introduced the following change:

Instead of previously stating:

“To address this issue, we aimed to rescue replicative stress in NONO and SFPQ depleted...”

We state now:

“To address this issue we aimed to rescue RPA32 phosphorylation levels in NONO and SFPQ depleted...”

Page 9, line 12:

Addressing the concern of reviewer 2 on the definition of replication stress we introduced the following change:

Instead of previously stating:

“...preventing RNA:DNA hybrid accumulation and R-loop related replicative stress at telomeres.”

We state now:

“...preventing RNA:DNA hybrid accumulation and R-loop related replication defects at telomeres.”

Page 9, line 19:

Addressing the concern of reviewer 2 on long-term experiments using NONO and SFPQ loss of function models, we highlighted that we are interested in addressing the direct impact of NONO and SFPQ on telomere function. We state now:

“In order to understand the direct importance of NONO and SFPQ for telomere integrity we performed short term loss of experiments using telomere chromosome orientation DNA-FISH (CO-FISH).”

Page 10, lines 1→:

Increasing the number of analysed telomeres in a new set of fragility experiments (revised Fig 4; revised Supplementary Fig. 4) we have obtained more precise and reliable information on the role of NONO and SFPQ in telomere fragility.

In the list below we want to demonstrate that the data of the original version of the manuscript were reproduced in the new set of experiments. Naturally, new experiments show improved p-values (please see respective figure panels). The text of revised version of the manuscript was changed in accordance to the new dataset.

NONO loss of function experiments:

U2OS cells (Fig. 4B):

- A. Telomere leading strand fragility (CCCTAA):
Original manuscript-version: +40%
Revised manuscript version: + 60%
- B. Telomere lagging strand fragility (TTAGGG):
Original manuscript-version: no significant alteration
Revised manuscript version: no significant alteration

H1299 cells (Supplementary figure 4A):

- A. Telomere leading strand fragility (CCCTAA):
Original manuscript-version: +47%
Revised manuscript version: + 59%
- B. Telomere lagging strand fragility (TTAGGG):
Original manuscript-version: +66%
Revised version: no significant alteration + 69%

NONO gain of function experiments:

U2OS cells (Fig. 4C):

- C. Telomere leading strand fragility (CCCTAA):
Original manuscript version: - 30%
Revised manuscript version: -26%
- D. Telomere lagging strand fragility (TTAGGG):
Original manuscript-version: no significant alteration
Revised manuscript version: no significant alteration

H1299 cells:

- C. Telomere leading strand fragility (CCCTAA):
Original manuscript-version: -39%
Revised manuscript version: -31%
- D. Telomere lagging strand fragility (TTAGGG):
Original manuscript-version: -45%
Revised version: no significant alteration: -33%

SFPQ loss of function experiments:

U2OS cells: (Fig. 4D)

- E. Telomere leading strand fragility (CCCTAA):
Original manuscript-version: -60%
Revised manuscript version: - 34%
- F. Telomere lagging strand fragility (TTAGGG):
Original manuscript-version: no significant alteration
Revised manuscript version: no significant alteration

H1299 cells:

- E. Telomere leading strand fragility (CCCTAA):
Original manuscript-version: -40%
Revised manuscript version: -38%
- F. Telomere lagging strand fragility (TTAGGG):
Original manuscript-version: no significant alteration
Revised manuscript version: no significant alteration

SFPQ gain of function experiments:

U2OS cells (Fig. 4E):

- G. Telomere leading strand fragility (CCCTAA):
Original manuscript-version: no significant alteration
Revised manuscript version: no significant alteration
- H. Telomere lagging strand fragility (TTAGGG):
Original manuscript-version: no significant alteration
Revised manuscript version: no significant alteration

H1299 cells:

- G. Telomere leading strand fragility (CCCTAA):
Original manuscript-version: no significant alteration
Revised manuscript version: no significant alteration
- H. Telomere lagging strand fragility (TTAGGG):
Original manuscript-version: no significant alteration
Revised manuscript version: no significant alteration

Based on these improved results we state now:

RNAi mediated depletion of NONO in U-2 OS cells significantly increased the appearance of aberrantly shaped or multi-dotted telomere signals the CCCTAA repeat containing telomeric strand in U-2 OS cells (+60%; Fig. 4B). Interestingly, loss of NONO does not have an impact on telomere fragility at the TTAGGG repeat containing lagging strand (Fig. 4B). This data is in line with leading strand fragility triggered by increased RNA:DNA hybrid abundance in RNaseH1 or Flap endonuclease loss of function cells^{17,36}. As expected, ectopic expression of NONO reduced basal levels of telomere lagging strand fragility (Fig. 4C). Remarkably, loss of NONO in telomerase positive cells results in telomeric leading and lagging strand fragility; accordingly, NONO overexpression reduces basic telomere fragility levels on both telomere strands (Supplementary figure 4A, B). Together, these data identify NONO as novel suppressor of telomere fragility that has a particular relevance in suppressing fragility at the telomeric leading strand in ALT cells that are reported to be prone to exhibit telomeric RNA:DNA hybrids and leading strand fragility.

Page 10:

Following the suggestion of reviewer 3 to control the conclusions of the experiments, we deleted the phrase:

Original version, Page 10, line 225: “This suggests that loss of SFPQ results in the release of mechanisms that may “repair” telomere fragility.

At this point of the manuscript this conclusion is not adequate.

Page 11, line 7:

We exchanged “10-fold increase” for “significant increase”

Page 11, line 15:

Increasing number of analysed telomeres in CO-FISH studies resulted in altered T-SCE values. Similar to fragility experiments we only observed a quantitative but not qualitative change of the results.

The old version states that:

“...co-depletion of SFPQ resulted in a dramatic, 60-fold increase of T-SCE of T-SCE that involves 60% of detectable telomeres.”

In the new version of the manuscript we state that:

“...co-depletion of SFPQ resulted in a dramatic, 18-fold increase of T-SCE of T-SCE that involves 35% of detectable telomeres.”

Page 11, line 17:

Following the suggestion of reviewer 3 to verify the conclusions stated in the manuscript, we introduced the following change:

Original version (line 250) “This suggests that telomere sister chromatid recombination represents an effective mechanism to resolve telomere fragility.

Revised version of manuscript:

“This suggests that telomere sister chromatid recombination may represents an effective mechanism to resolve telomere fragility triggered by NONO depletion.”

Page 11, line 20:

We inserted new text related to the new Figure 5I J and new Supplementary figure 5A, B:

“In line with increased T-SCE frequency in NONO/SFPQ depleted cells we observed an increased frequency of co-localization of TRF2 with Promyelocytic Leukemia (PML) nuclear bodies and co-localization of RAD51 with TRF2 in U-2 OS cells that were depleted for SFPQ (Fig. 5G-J). Accordingly, we found significantly elevated TERRA – PML co-localization in SFPQ knock-down cells, a feature reported for ALT cells (Ref. 17; Supplementary figure 5A, B).

Page 12, line 4:

Following the suggestion of reviewer 3 to verify the conclusions stated in the manuscript, we introduced the following change:

Original version (line 257): “Depletion of NONO does not impact on APB frequency, underlining that loss of SFPQ is the major trigger for the enhancement of the ALT pathway in U-2 OS cells.”

Revised version of manuscript:

“As expected, depletion of NONO does not impact on APB frequency in U-2 OS cells.”

Page 12, line 5:

We removed the redundant Supplementary figure 5A, B of the original version of the manuscript (shows same like Fig. 6C,D in the old version of the manuscript). We give reference to the revised Fig. 6C, D.

We therefore state in the revised version of the manuscript:

“...depletion of SFPQ in telomerase positive H1299 cells triggers T-SCE we did not observe significantly increased APB numbers (Fig. 6C, D)”

Page 13:

Increasing the number of analysed telomeres in telomere length measurements we have obtained more precise and reliable information on the role of NONO and SFPQ in telomere length control. As a consequence, we needed to exchange the respective values (%) for telomere length alterations.

Page 13, line 3: siNONO in U2OS: +8% instead of +17% telomere length increase; siSFPQ in U2OS; +16% instead of +20% telomere length increase.

Page 13, line 5: combined knock-down in U2OS: +35% instead of +30%

Page 13, line 8: in H1299 knock-down of NONO/SFPQ: -21% reduction instead of -10%/-18% reduction.

Page 13, line 23: increased telomere length 5-Aza-2'-deoxycytidine treated SFPQ knock-down H1299 cells: +47% instead of +29%.

Discussion Section:

Page 14, line 19:

Addressing the concern of reviewer 3 on the identity of telomere repeats bound by NONO/SFPQ we exchanged “novel telomere associated proteins” for “novel telomere repeat associated proteins”

Page 14, line 24: To address a comment by reviewer 3, we added a comment that introduces the possibility that ECTRs may also be associated with SFPQ/NONO. “In ALT cells, extra-chromosomal telomeric repeats (ECTR) can constitute a notable fraction of the total telomere repeat content, suggesting that a fraction of NONO and SFPQ may also locate to this type of telomere repeat sequences⁵¹. However, alterations at telomeres of metaphase chromosomes observed in loss of function experiments support a role for NONO and SFPQ at telomeres.”

Page 15, line 9:

In order to address concerns of reviewer 2 on replication stress markers, we have exchanged the following sentence:

Original version of manuscript:

“In line with the induction of replication stress we observed increased loading of the classic DNA damage marker γ H2AX at telomeres of NONO or SFPQ depleted cells.”

Revised version of manuscript:

“In line with the increased abundance of RNA:DNA hybrids we observed increased loading of the classic DNA damage marker γ H2AX at telomere repeats of NONO or SFPQ depleted cells.”

Page 15, line 17:

Following the suggestion of reviewer 3 to verify the conclusions stated in the manuscript, we introduced the following change:

Original version of manuscript:

“NONO appears to have a selective role in preventing fragility of the leading, CCCTAA repeat containing telomeric strand

Revised version of manuscript:

“NONO appears to play an important role in preventing fragility of the leading, CCCTAA repeat containing telomeric strand”

Page 15

We deleted the phrase: “thus providing additional support for a role of NONO in suppressing R-loop formation” because it only contains redundant information.

Page 15, line 22:

In order to address concerns of reviewer 2 on the definition of the term “replication stress” we have exchanged the words “replicative stress” (original version) for “impaired replication” (revised version)

Page 15, line 24

Following the suggestion of reviewer 3 to verify the conclusions stated in the manuscript, we introduced the following change:

Original version of manuscript:

“This effect is paralleled by a reduction of telomere fragility, indicating that homologous recombination can rescue telomere fragility. This indicates that SFPQ functions as an important barrier to homologous recombination at telomeres.”

Revised version of manuscript:

“This effect is paralleled by a reduction of telomere fragility, leading to the interesting speculation that homologous recombination may rescue telomere fragility. Our data show that SFPQ functions as barrier to homologous recombination at telomeres”

Page 16, line 3

We have increasing the number of analysed telomeres in T-SCE studies, resulting different T-SCE values (but still preserving the original “qualitative” biological data).

We exchanged the T-SCE value of the original version of the manuscript (60%) for the value of the new T-SCE results (35%).

Page 16, line 10

Addressing the concern of reviewer 2 on long-term experiments using NONO and SFPQ loss of function models, we highlighted that we are interested in addressing the direct impact of NONO and SFPQ on telomere function. We state now:

“Thus, NONO and SFPQ have a direct role in suppressing RNA:DNA hybrid related telomere fragility and recombination (Fig. 7)”

Page 16, line 12

We included the results from **new Fig 5I, J** into the discussion section and state now:

“In line with this, we found that NONO/SFPQ knock-down in U-2 OS triggers increased T-SCE frequencies, increased localization of RAD51 at telomere repeats as well as the engagement of telomeres in APBs, thus resulting telomere elongation”

Page 16, line 21

Following the suggestion of reviewer 3 to verify the conclusions stated in the manuscript, we modified the following phrase by replacing “are required” to “are important”:

“This suggests that functional APBs are important to convert increased T-SCE frequencies in SFPQ loss of function cells into an overall increase in telomere length”.

Figure legends

Page 24, legend to Fig. 1:

Description of ChIP dot blot diagram (**revised panel G**) was inserted

Page 24, legend to Fig. 2:

Description of box blot diagram of **revised panel C** was inserted

Page 25, legend to Fig. 3:

Figure legend related to panel G of the original version of the manuscript was removed

Page 26, legend to Fig. 4:

Description of box blot diagram for **revised panels B, C, D, E** was inserted

Page 26, legend to Fig. 5:

Description of **new Fig. 5I, J** was inserted

Description of box blot diagrams of **revised panels B, C, E, F** was inserted

Page 27, legend to Fig. 6:

Description of box blot diagram of **revised Fig. 6** was inserted

Supplementary material and methods:

Page 4, Microscopy:

In the revised version of the manuscript, details on analysed focal planes are indicated

Supplementary figure legends:

Page 7, Supplementary figure 1

Description of **new figure panels C-F**. was inserted.

Page 7, Supplementary figure 2

Description of **new figure panel D** was inserted.

Description of **new figure panel G, H** and description of **new box blot diagrams (F, H)** was inserted.

Description of **new figure panel I, J** was inserted

Page 7, Supplementary figure 3

Figure legend for **panel E** was removed.

Page 8,9 Supplementary figure 4

Description of box blot diagram for **revised panels A-D** was inserted

Page 9, Supplementary figure 5

Note: Supplementary figure 5 of the original version (PML-TRF2 co-localization) was replaced for TERRA-PML co-localization data. Accordingly, also the respective figure legend has been replaced **(new Supplementary figure 5)**.

Figure 1 for Reviewers. Confocal microscopy analysis of SFPQ and NONO localization at telomeres. A. Immunofluorescence with anti-SFPQ and anti-TRF1 antibodies was performed in U2OS cells and Z-stack images were acquired on confocal microscopy. Representative images show one focal plane. **B.** SFPQ-TRF1 co-localizations were scored and counted by using colocalization plugin of ImageJ software. Mean number of co-localizations events in perfect match of the two channel (-) or after shift of SFPQ channel (+) is shown in the graph. **C.** Immunofluorescence with anti-NONO and anti-TRF2 antibodies was performed in U2OS and Z-stack images were acquired on confocal microscopy. Representative images show one focal plane. **D.** NONO-TRF2 co-localizations were scored and counted by using colocalization plugin of ImageJ software. Mean number of co-localizations events in perfect match of the two channel (-) or after shift of NONO channel (+) is shown. Error bars indicate standard deviation. N= number of independent experiments. n= number of analyzed nuclei. Unpaired t-test was used to calculate statistical significance; p-values are shown.

We found that shifting one channel versus the other strongly reduced TRF1-SFPQ colocalization and also significantly reduced TRF2-NONO co-localization in U2OS interphase cells. This data support that a significant fraction of nuclear NONO and SFPQ localizes to telomeres in interphase cells. This data has not been included into the revised version of the manuscript and is only accessible to the reviewers and the editor of the manuscript.

In contrast to the strong reduction of colocalization observed TRF1-SFPQ channel shifting (ca. -3fold, $p < 0,009$), shifting of channels in TRF2-NONO co-localization experiments causes a less dramatic reduction of the co-localization index (-18%, $p < 0,034$). We think that this is mainly caused by the different numbers of NONO and SFPQ foci observed in our immunolocalization studies. Anti-NONO antibodies reveal a much higher number of NONO foci when compared to the number of SFPQ foci per nucleus (Fig. 1D, E; Supplementary figure 1E, F). Consequently, the chance of having random TRF2-NONO colocalization after channel shifting is much higher compared to TRF1-SFPQ channel shifting experiments.

We want to underline that Chip experiments support evidence for a localization of NONO at telomere repeat sequences (Fig. 1G). Finally, immunoprecipitation experiments show interaction of NONO with TRF1 and TRF2 (Supplementary fig. 1C,D)

Figure 2 for Reviewers. A. Western blotting of whole cell extracts derived from U-2 OS cells transfected with indicated siRNAs. Extracts were prepared 24 hrs post-transfection. A set of surrogate markers were used to monitor activation of replication defects. Values of quantitative densitometric analysis are reported on the top of each blot. si-Control was set to 1. Treatment with 5 mM hydroxyurea (HU) for 6 hrs was used as control for DNA damage response activation.

B. Quantification of (A). Expression of shown markers were quantified against tubulin. In the graph ratios between p-ATR/ATR are shown. Chk1 levels and phosphorylation of RPA32Ser33 was quantified against tubulin.

C. Quantification of pan-nuclear RPA32pSer33 fluorescence intensity (left) and p-ATR (right) in U-2 OS cells transfected with indicated siRNAs. For quantification, images from Figure 3 were used. N=number of independent experiments. n= number of analysed nuclei. A Student's t-test was used to calculate statistical significance; p-values are shown.

In the manuscript we use IF to show increased phospho-RPA32Ser33, gammaH2AX and phospho-ATR at telomere repeat sequences of experimental cells (**revised Fig. 3, revised Supplementary fig. 2I, J**). Western blotting revealed phosphorylation of gammaH2AX after a knock-down of SFPQ/NONO for 3 days (**revised Supplementary Figure I, J**). At this time point we did not obtain a convincing increase in ATR phosphorylation (comment by reviewers, original version of the manuscript). We have performed time-course knock-down experiments to better follow the kinetics of the phosphorylation of ATR, Chk1 and RPA32Ser33. Western blotting after 24 hours of NONO/SFPQ knock-down revealed increase of phosphorylation of ATR in NONO and SFPQ depleted cells and a modest increase of phosphorylated RPA32Ser33 in SFPQ depleted cells (A,B). The newly ordered phospho-Chk1 antibody did not give results in western blotting (and immunofluorescence) experiments, presumably for technical reasons related to the antibody-batch.

To further investigate this issue, we used immunofluorescence images from Fig. 3 and quantified pan-nuclear phospho-ATR and pan-nuclear phosph-RPA32ser33 levels using ImageJ. Importantly, we found a significant increase of pan-nuclear RPA32Ser33 phosphorylation upon NONO or SFPQ depletion (C, left panel). Loss of SFPQ further revealed an increase in pan-nuclear phospho-ATR levels (C, right panel). In fact, results from telomeres (**Fig. 3**) are reproduced on the global level in experimental cells. We are confident that these results support the activation of surrogate makers of replication stress.

Our results suggest that IF is superior to western blotting in detecting p-ATR and p-RPA32Ser33. Data from gammaH2AX experiments indicate that loss of NONO or SFPQ does not induce a full-blown DNA damage response (**revised Supplementary fig. 2I, J**). We therefore propose that manipulation during extract preparation and western blotting results a significant reduction of phosphorylation of the respective targets, rendering difficult of detect an increase in pATR and/or p-RPA32Ser33. PFA fixation during IF appears to be a more adapted strategy to immediately block potential phosphatases and to maintain alterations in p-ATR and p-RPA32Ser33 levels.

Due to i) differences observed between western blotting and immunofluorescence and ii) the fact that new western data on RPA32Ser33 and ATR were obtained at a different timepoint (24hours) than gammaH2AX (72 hours knock-down) we decided to only include gammaH2AX western data in the revised version of the manuscript and provide RPA32Ser33 and ATR data to reviewers. We feel that this supports the clearness of the manuscript.

REVIEWERS' COMMENTS:

Reviewer #2 (Remarks to the Author):

The authors are to be commended for their effort in revision. They have satisfactorily addressed all issues systematically and thoroughly.

Minor note related to my initial comments regarding "Data presentation". These changes are not required for publication - my comments are to clarify for the authors' benefit.

I was suggesting that instead of displaying bar graphs, to visualize the data as dot plots (i.e. a single dot representing the mean of each replicate, and a line and whiskers for the mean \pm SD). This is a more accurate representation of the distribution of data which is hidden in bar graphs. This is also the current preferred data presentation. For more discussion see <https://doi.org/10.1371/journal.pbio.1002128>

For the dense dot plots I suggested box and whisker plots. This is because of the density of dots in your figures often the underlying data distribution. This density of data points is often resolved in box and whisker plots (violin plots also work). However in your current data sets many graphs still suffer from a very high density of outlying points (e.g. figure 6). It might be useful to the reader if the size of each data point were reduced. For example if you are using GraphPad Prism, reduce the pt size of data points to 1. This is simply to better visualize the data and ease reader comprehension.

Reviewer #3 (Remarks to the Author):

I appreciate the enormous work the authors have put into this manuscript and into addressing concerns. Especially the improvement of statistics was extremely important. I therefore support publication now.

REPLY TO REVIEWERS' COMMENTS

Reviewer #2 (Remarks to the author)

Comment by Reviewer 2

The authors are to be commended for their effort in revision. They have satisfactorily addressed all issues systematically and thoroughly.

Reply to Reviewer 2

We thank reviewer 2 for the positive evaluation of the revised version of our manuscript.

Comment by Reviewer 2

Minor note related to my initial comments regarding "Data presentation". These changes are not required for publication - my comments are to clarify for the authors' benefit.

I was suggesting that instead of displaying bar graphs, to visualize the data as dot plots (i.e. a single dot representing the mean of each replicate, and a line and whiskers for the mean \pm SD). This is a more accurate representation of the distribution of data which is hidden in bar graphs. This is also the current preferred data presentation. For more discussion see <https://doi.org/10.1371/journal.pbio.1002128>

Reply to Reviewer 3

We agree that data presentation in a dot-blot format would be preferable. After experimenting with different layouts, we felt that dot-blot representation makes only sense when the "control sample dot" for each experimental replicate is connected by a line with the respective "knock-down sample dot".

However, the situation is complicated by the fact that our graphs contain more than 2 experimental conditions. A typical example is Figure 2I where we have data from siControl, siRNaseHI, siNONO and siSFPQ experiments (all $n=3$).

Using lines to interconnect the siControl values of three replicates ($n=3$) with the three replicates of each knock-down experiment would not generate a clear data representation. Showing only data points is also not a good option for presenting our data in a clear way.

For this reason, we would prefer to not change the data presentation of experiments with low number of data-points. We hope that reviewer 3 can agree to this decision

A representative image provided as Figure 1 to the reviewers may support our decision.

Comment by Reviewer 2

For the dense dot plots I suggested box and whisker plots. This is because of the density of dots in your figures often the underlying data distribution. This density of data points is often resolved in box and whisker plots (violin plots also work). However, in your current data sets many graphs still suffer from a very high density of outlying points (e.g. figure 6). It might be useful to the reader if the size of each data point were reduced. For example, if you are using GraphPad Prism, reduce the pt size of data points to 1. This is simply to better visualize the data and ease reader comprehension.

Reply to Reviewer 2

We have modified the data-representation, as requested by reviewer 2. Quantification of TERRA RNA-FISH and telomere DNA-FISH data was modified in order to better visualize the distribution of data points in **Figure 2C** and **Figure 6B, D, H**.

As representative example that compares the new layout with the old layout of Figure 6B (Q-FISH data) is provided as *Figure 2 to the reviewers*.

Reviewer #3 (Remarks to the author)

Comment by Reviewer 3

I appreciate the enormous work the authors have put into this manuscript and into addressing concerns. Especially the improvement of statistics was extremely important. I therefore support publication now.

Reply to Reviewer 3:

We thank reviewer 3 for the positive evaluation of the revised version of our manuscript.